# Dialogue Without Limits: Constant-Sized KV Caches for Extended Response in LLMs

**Ravi Ghadia** [1]   **Avinash Kumar** [1]   **Gaurav Jain** [2]   **Prashant Nair** [3]   **Poulami Das** [1]

## Abstract

Autoregressive Transformers rely on Key-Value (KV) caching to accelerate inference. However, the linear growth of the KV cache with context length leads to excessive memory consumption and bandwidth constraints. Existing methods drop distant tokens or compress states in a lossy manner, sacrificing accuracy by discarding vital context or introducing bias.

We propose MorphKV, an inference-time technique that maintains a constant-sized KV cache while preserving accuracy. MorphKV balances long-range dependencies and local coherence during text generation. It eliminates early-token bias while retaining high-fidelity context by adaptively ranking tokens through correlation-aware selection. Unlike heuristic retention or lossy compression, MorphKV iteratively refines the KV cache via lightweight updates guided by attention patterns of recent tokens. This approach captures inter-token correlation with greater accuracy, which is crucial for tasks like content creation and code generation. Our studies on long-response tasks show 52.9% memory savings and 18.2% higher accuracy on average compared to state-of-the-art prior works, enabling efficient deployment.

## 1. Introduction

Large Language Models (LLMs) have become indispensable for tasks requiring extensive context retention (e.g., document summarization) and prolonged text generation (e.g., code synthesis). As model architectures become sophisticated, their ability to process nuanced inputs and produce coherent, long-form outputs has improved dramatically.

[1]Department of Electrical and Computer Engineering, University of Texas at Austin, Texas, USA [2]D-Matrix [3]University of British Columbia, USA. Correspondence to: Ravi Ghadia <rghadia@utexas.edu>.

*Proceedings of the 42nd International Conference on Machine Learning*, Vancouver, Canada. PMLR 267, 2025. Copyright 2025 by the author(s).

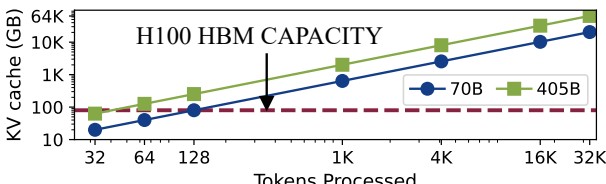

*Figure 1.* KV cache sizes for the Llama 3.1 70B and 405B models across varying sequence lengths with a batch size of 256.

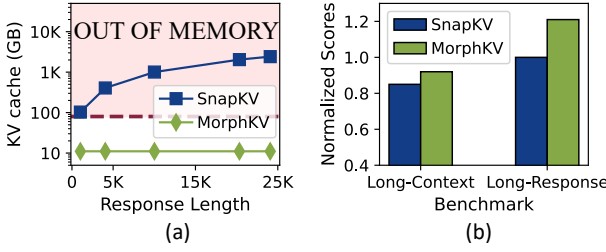

*Figure 2.* (a) Despite compression, the state-of-the-art SnapKV memory footprint increases with response length and exceeds available HBM capacity even on high-end GPUs. This study uses the Qwen 2.5 7B model on an NVIDIA H100 and the Long-Writer benchmark. (b) Even at lower memory capacity, MorphKV achieves higher accuracy than SnapKV for long-response tasks.

However, this progress is hindered by the memory overhead of Key-Value (KV) caches. KV caches store the key-value pairs to enable attention mechanisms for auto-regressive decoding during LLM inference. Unfortunately, as shown in Figure 1, the KV cache size grows with sequence length, often exceeding the memory capacity of even high-end GPUs.

The distinction between long-context and long-response tasks lies in the phase where token processing dominates. Long-context tasks, such as document summarization and prompt comprehension, primarily process a large volume of input tokens during the prefill phase, where the model ingests and encodes the initial prompt. In contrast, long-response tasks, such as essay writing and code generation, generate a substantial number of output tokens during the decode phase, requiring sustained attention over growing sequences of self-generated tokens.

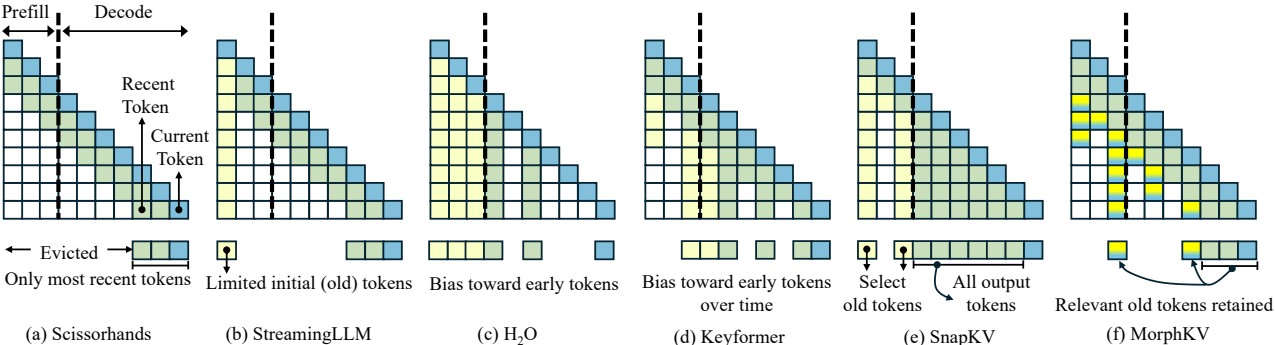

*Figure 3.* Illustrative comparison of KV cache reduction methods as tokens are processed. (a) Scissorhands retains only a window of *recent* (shown in green) tokens, (b) StreamingLLM also stores a few initial tokens (*old* shown in yellow) from the prefill step, and (c) $H_2O$ stores even more old tokens (all prompt tokens) and only relevant recent tokens. (d) Keyformer stores only important old and recent tokens but remains biased towards the early tokens. (e) SnapKV retains selected old tokens from prefill and all decode (more recent) tokens (f) MorphKV identifies and stores only those old tokens that correlate with recent tokens.

Numerous approaches have been proposed in the literature to minimize the impact of growing KV cache sizes. In studies like FlashAttention (Dao, 2023) and vLLM (Kwon et al., 2023), the authors propose techniques to either materialize only partial caches at a time or use paging techniques by fragmenting KV caches into smaller blocks, thereby avoiding the need to reserve memory for the entire cache at once. Beyond this, prior works like FastGen (Ge et al., 2023) and MiKV (Yang et al., 2024) compress KV caches by retaining only a subset of KV pairs from recent and older tokens, prioritizing those deemed important based on attention scores and discarding the rest. However, this creates a *trade-off*: while memory savings increase as more KVs are discarded, the accuracy depends on the retained KVs effectively capturing context for future tokens. Consequently, these methods often sacrifice accuracy for reduced memory usage.

For example, as shown in Figure 3, Scissorhands (Liu et al., 2024b) retains only the KVs of recent tokens, sacrificing accuracy by discarding past context. StreamingLLM (Xiao et al., 2023) improves accuracy slightly by preserving KVs of a few initial tokens (attention sinks) alongside recent tokens but struggles when early tokens fail to capture sufficient context. $H_2O$ (Zhang et al., 2023) retains KVs from the entire input prompt and the most attended output tokens, achieving high performance but reduced memory savings due to the large number of retained KVs. It also suffers from selection bias during decoding, preserving unimportant past KVs, which hinders performance in long-response tasks.

Keyformer (Adnan et al., 2024) selects top old and recent tokens for memory savings and uses Gumbel noise to reduce selection bias. While somewhat effective, it cannot entirely eliminate selection bias because Gumbel noise by itself introduces new forms of biases (Mussmann et al., 2017), reducing accuracy for long-context and long-response tasks. SnapKV (Li et al., 2024), the current state-

of-the-art, achieves high accuracy in long-context tasks by retaining the most attended tokens from the input prompt. However, as shown in Figure 2(a), it retains *all* generated output tokens from the decode phase, causing the KV cache size to scale with response lengths, making it unsuitable for long-response tasks. Overcoming these limitations is crucial for improving LLM performance in applications like scripting and content creation (long-response tasks).

**Our Proposal – *MorphKV*:** *MorphKV* achieves a constant-size KV cache by retaining only a limited number of old and recent tokens. However, to achieve higher accuracy, *MorphKV* employs a more dynamic KV selection algorithm that analyzes the attention patterns of the current token toward retained KVs. Unlike prior methods that independently identify important tokens, *MorphKV* retains only those old tokens that correlate strongly with recent tokens.

To better capture context, MorphKV prioritizes the attention scores of relevant recent tokens rather than relying on historically most-attended tokens, addressing bias issues observed in methods like $H_2O$ and Keyformer. As shown in Figure 2(b), MorphKV achieves better scores than SnapKV: for Phi4, up to 8% higher for long-context task (VCSum) while saving 56% on KV cache memory, and for Qwen2.5, up to 21% higher score for the long-response task (LongGen-Bench) while saving 83% on KV cache memory. This shows the impact of retaining a compact set of high-quality KVs and an improved attention mechanism in MorphKV.

MorphKV improves accuracy by 9.4% and 18.2% on average compared to SnapKV and $H_2O$ while reducing the KV cache footprint by 88.1% and 52.9% respectively for long-response tasks.

MorphKV is now open-source, and accessible at `https://github.com/ghadiaravi13/MorphKV`.

## 2. Background and Motivation

### 2.1. Large Language Model Inference

LLM inference begins with the *prefill* step, where the model processes the input prompt and generates *Key-Value* (KV) pairs for each token in the prompt. Next, in the *decode* phase, the model generates output tokens auto-regressively such that each output token attends to the KV pairs of all preceding tokens in the sequence while creating its own KV pair. This *attention* mechanism enables the LLM to maintain context and produce coherent responses. The KV pairs are stored in memory structures known as *KV caches*.

KV caches scale with the number of tokens processed, becoming prohibitively large for long-context and long-response tasks and posing significant challenges in deploying LLMs. Long-context tasks, such as creating diet plans from medical histories or summarizing documents like manuals, loan agreements, or papers, involve long prompts with many input tokens. In contrast, long-response tasks such as crafting lesson plans, providing step-by-step instructions, or writing scripts generate numerous output tokens from short inputs. While both types of tasks require large KV caches, they differ in when the KVs are produced. Long-context tasks generate most KVs in the prefill step, unlike long-response tasks that create most KVs during decoding.

### 2.2. Limitations of KV Cache Compression Methods

KV cache compression addresses their growing memory footprint through several strategies, such as quantization, algorithmic optimizations, cross-layer and cross-head approaches, and pruning. Quantization-based methods store KV pairs using lower precision (Kang et al., 2024; Zhang et al., 2024), whereas algorithmic methods modify attention-layer computations (Chang et al., 2024; Saxena et al., 2024). Cross-layer optimizations leverage inter-layer similarities, selectively retaining KVs from layers with significant contributions (Yuan et al., 2024; Saxena et al., 2024; Cai et al., 2024). Cross-head optimization (Fu et al., 2024; Feng et al., 2024) reduces the KV cache footprint by retaining KVs only from the most impactful attention heads, as different heads contribute unevenly to model performance. Pruning strategies selectively retain a subset of *important* KV pairs (Adnan et al., 2024; Xiao et al., 2023; Li et al., 2024; Zhang et al., 2023; Liu et al., 2024b) and are more effective than other approaches because they capture task-specific context more accurately.

However, compressed KV caches are either limited by accuracy or scalability. Constant-sized KV caching methods, such as Scissorhands, StreamingLLM, and Keyformer, have limited accuracy. On the other hand, more accurate methods like SnapKV are not scalable as they fail to address the growing KV cache size for long-response tasks.

## 3. MorphKV

This paper proposes *MorphKV*, a KV compression technique that reduces the KV cache size without compromising accuracy. MorphKV partitions the context into two components: *recent context* ($\mathcal{R}$) and *distant context* ($\mathcal{D}$). The recent context $\mathcal{R}$ corresponds to the last $R$ tokens that preserve local coherence, while the distant context $\mathcal{D}$ captures long-range dependencies. By attending to both $\mathcal{R}$ and a selective subset of $\mathcal{D}$, MorphKV ensures that the generated text remains contextually coherent and semantically meaningful.

Figure 4 presents an overview of our proposed design. A key insight in MorphKV is that tokens in $\mathcal{R}$ have already attended to tokens in $\mathcal{D}$ during their generation. Therefore, rather than retaining all or a subset of older tokens based on aggregated patterns, MorphKV leverages the attention profiles of recent tokens to select only the most relevant distant tokens. In this way, MorphKV constructs a compact yet accurate KV cache of size $C + R$, where $C$ is the number of distant tokens retained and $R$ is the number of recent tokens. Specifically, MorphKV ❶ ranks older tokens based on their relevance to the recent tokens using a specialized algorithm that performs element-wise transformations, denoted by $f(x)$ in Figure 4. As attention scores inherently quantify how strongly past tokens were attended to during prior generations, using the attention scores of the recent tokens helps surface the most contextually relevant older tokens. Next, MorphKV ❷ selectively retains only the most correlated old tokens in the KV cache, evicting those deemed irrelevant. This approach ensures an optimized memory footprint while preserving essential long-range dependencies.

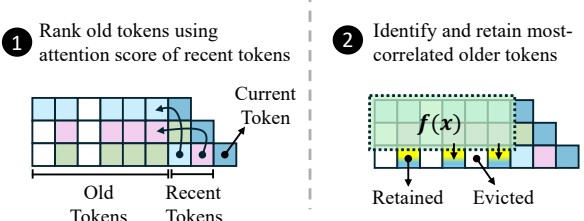

*Figure 4.* Overview of MorphKV. (1) MorphKV uses the most recent window tokens to capture neighboring context and their attention scores to rank the older tokens. (2) To capture relevant distant context, MorphKV only retains old tokens maximally correlated to the recent window tokens by consulting the attention scores aggregated using a fusion function $f(x)$.

### 3.1. Mathematical Formulation

Let $Q_i$, $K_i$, and $V_i$ be the query, key, and value vectors for the token being generated at timestep $i$. Let $G_i$ denote the KV cache storing $(K_j, V_j)$ pairs for all the previously generated tokens $j < i$. The standard attention mechanism (Vaswani et al., 2023) computes the attention weights

$AW_i$, as shown in Equation (1):

$$AW_i = \text{Softmax}\left(\frac{Q_i K^T}{\sqrt{d_h}}\right), \quad O_i = AW_i \cdot V, \quad (1)$$

where $K = [K_1, K_2, \ldots, K_{i-1}] \in \mathbb{R}^{(i-1) \times d_h}$, $V = [V_1, V_2, \ldots, V_{i-1}] \in \mathbb{R}^{(i-1) \times d_h}$, and $d_h$ is the hidden dimension. The attention output $O_i \in \mathbb{R}^{d_h}$ encodes the information from all the previous tokens.

### 3.1.1. PROBLEM STATEMENT

Retaining every key-value pair $(K_j, V_j)$ for $j < i$ can increase memory usage as $i$ grows. Let $G_i^*$ be an *optimal* reduced cache of size $C + R$ that minimizes the change in the attention output as denoted in Equation (2):

$$G_i^* = \arg \min_{\substack{G_i' \subseteq G_i \\ |G_i'| = C+R}} \| O_i - O_i' \|_2, \quad (2)$$

where $O_i'$ is the attention output, as shown in Equation 1, computed using only the tokens in $G_i'$. We want $\| O_i - O_i' \|_2 \leq \epsilon$ for a small $\epsilon \geq 0$, ensuring minimal error despite reducing the KV cache to $C + R$ entries.

### 3.1.2. APPROXIMATING OPTIMAL KV CACHE

Although solving Equation 2 directly is intractable because of its combinatorial nature, MorphKV adopts two intuitive heuristics (H1 and H2):

- **Local Coherence (H1):** Always retain the last $R$ tokens to preserve continuity.

- **Distant Relevance (H2):** Retain only the $C$ *most informative* older tokens, as measured by *fused* attention scores with respect to the $R$ recent tokens.

Concretely, we define an approximate policy $\mathcal{P}'$ in Equation (3) as shown below:

$$G_i' = \mathcal{P}'(G_i, F_i), \quad \text{where } |G_i'| = C + R. \quad (3)$$

Here, $G_i' \subseteq G_i$ contains (1) the $R$ most recent entries $G_i^R$ and (2) the top-$C$ older entries selected based on an auxiliary score vector $F_i$.

### 3.1.3. DEVELOPING THE AUXILIARY SCORE VECTOR $F_i$

Let $\mathcal{W}_i = \{w_{i-1}, w_{i-2}, \ldots, w_{i-R}\}$ denote the window of $R$ most recent tokens at timestep $i$. MorphKV inspects the attention weights of these tokens to build $F_i$, using Equation (4). Specifically,

$$F_i[k] = f\left(AW_{i-1}[k], AW_{i-2}[k], .., AW_{i-R}[k]\right) \quad (4)$$

where $AW_r[k]$ is the attention weight that $w_r$ assigned to the $k$-th older token, and $f(\cdot)$ is a *fusion function*. For $f(\cdot)$,

MorphKV proposes two choices, the sum and max fusions, as shown in Equations (5) and (6) respectively:

$$\text{Sum Fusion: } F_i[k] = \sum_{r=i-1}^{i-R} AW_r[k] \quad (5)$$

$$\text{Max Fusion: } F_i[k] = \max_{r=i-1}^{i-R} AW_r[k]. \quad (6)$$

The **Sum Fusion** prefers tokens consistently attended to across multiple recent tokens, whereas the **Max Fusion** selects tokens strongly attended by at least one recent token. The intuition is that tokens frequently or strongly attended to by recent tokens are likely critical for maintaining long-range coherence. For example, recurring entities, such as characters in a story, often receive sustained attention across multiple decoding steps. Algorithm 1 shows the dynamic token selection process using the auxiliary score vector.

---

**Algorithm 1** MorphKV: Dynamic Token Selection

**Input:**

- KV cache $G_i$ (keys/values of older tokens)

- Tokens $x_{1:i}$ (sequence generated so far)

- Num of Query-Heads, Key-Heads: $\{M, M'\}$

- Per-head attention weights $\{AW^m\}_{m=1}^M$

- Window size $R$, fusion function $f$, capacity $C$

**Output:** Updated cache $G_{i+1}$

$\mathcal{W}_i \leftarrow \{x_{i-1}, x_{i-2}, \ldots, x_{i-R}\}$     ▷ Recent tokens
**for** $w_r \in \mathcal{W}_i$ **do**
    $S_r \leftarrow \sum_{m=1}^{(M/M')} AW_r^m$   ▷ Aggregate scores if GQA
**end for**
$F_i \leftarrow f(S_{i-1}, S_{i-2}, \ldots, S_{i-R})$    ▷ Fuse recent tokens' scores
$G_{i+1} \leftarrow \text{Top}_C(F_i) \cup \mathcal{W}_i$   ▷ Retain top-$C$ distant + $R$ recent
**Return:** $G_{i+1}$

---

### 3.1.4. SELECTION OF KV PAIRS

After computing $F_i$, we pick the top-$C$ entries (older tokens) according to $F_i$ and combine them with the $R$ most recent tokens, as shown in Equation (7):

$$G_{i+1} = \left\{\text{Top}_C(F_i)\right\} \cup \left\{R \text{ recent tokens}\right\} \quad (7)$$

Hence, $G_{i+1}$ contains $C + R$ tokens in total, satisfying heuristics (H1) and (H2). By updating $G_i \to G_{i+1}$ at each timestep, MorphKV prunes the KV cache incrementally, ensuring that memory usage remains fixed at $C + R$ while preserving essential local and distant context.

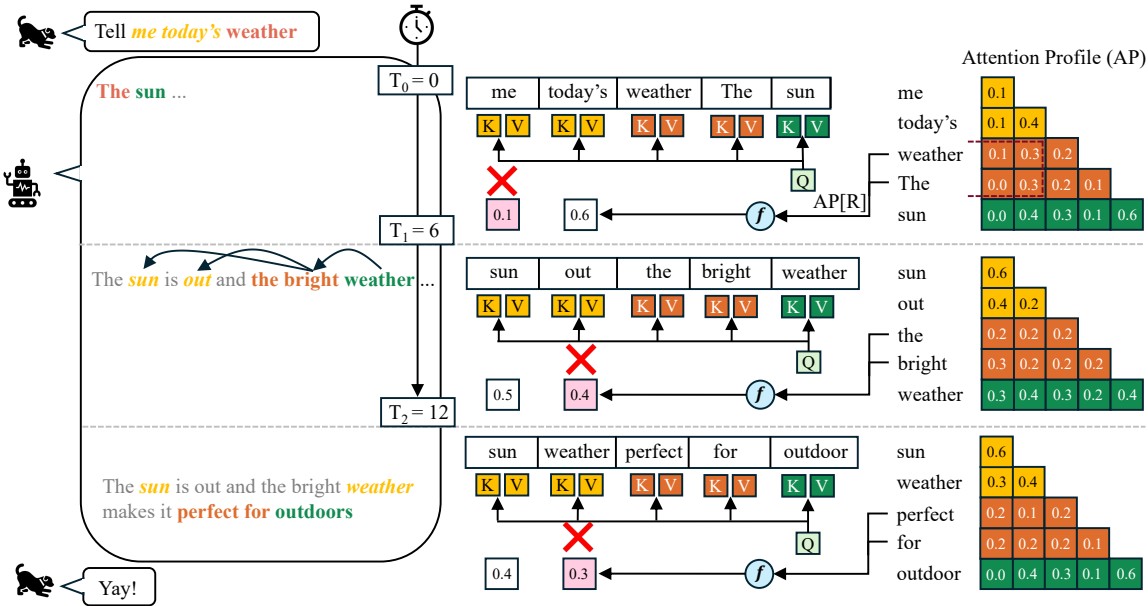

*Figure 5.* Illustration of the Key Value (KV) caching mechanism in MorphKV. MorphKV uses the insight that recent tokens naturally capture some distant context from old tokens due to the auto-regressive nature of token generation. For example, at decoding step $T_0$, MorphKV consults the Attention Profile of recent tokens *weather* and *The* to learn that these tokens have attended considerably more to the old token *today's* than *me*. MorphKV uses this information and evicts the latter.

### 3.2. Intuition with A Walk-Through Example

Figure 5 demonstrates the operations in MorphKV with $C = 2$ and $R = 2$, using sum fusion as the fusion function.

**At timestep $T_0$:** Suppose the recent tokens are [*weather*, *The*], while the old tokens are [*me*, *today's*]. The attention profiles (AP) of *weather* and *The* both strongly point to *today's* (e.g., 0.3 each) and weakly to *me* (e.g., 0.05 each). By summing these attention scores, MorphKV deems *today's* to be more relevant (score 0.6) than *me* (score 0.1). Consequently, it retains *today's* in $\mathcal{D}$ and evicts *me*.

**At timestep $T_1$:** The most recent tokens shift to [*perfect*, *for*], while older tokens include [*sun*, *out*]. MorphKV recalculates the fused attention scores for *sun* and *out*. If *out* receives a lower combined score than *sun* (as illustrated), it is evicted, preserving only *sun* in the distant context.

**At timestep $T_2$:** The most recent tokens are now [*for*, *bright*], and the older tokens still include *sun*. If *sun* continues to receive relatively high attention from the new recent tokens, it remains in the cache despite being one of the oldest tokens. This shows how MorphKV can preserve distant tokens of continued importance (e.g., *sun*) while evicting those that have very likely become less relevant over time.

### 3.3. Handling Multiple Heads

For transformer-based LLMs, attention is performed across multiple heads, called Multi-Headed Attention (MHA)

(Vaswani et al., 2023). In MHA with $M$ heads, each head maintains a separate KV cache, resulting in $M$ distinct caches and significant memory overhead. Modern LLMs like Llama3.1 (Dubey et al., 2024) use Grouped-Query Attention (GQA) (Ainslie et al., 2023) with $M'$ grouped-key heads ($M' < M$), where every $M/M'$ query heads share a single key/value head, reducing the number of KV caches to $M'$ and cutting memory usage by a factor of $M/M'$.

MorphKV is compatible with both MHA and GQA architectures. For MHA, we independently apply MorphKV to each of the $M$ heads, pruning their caches in parallel. For GQA, we first aggregate attention weights from the $M/M'$ grouped query heads, then compute fusion scores $F_i$ for the shared key-value heads. Our experiments show that MorphKV performs equally well with both approaches. By default, we choose GQA due to its memory efficiency. This flexibility distinguishes MorphKV from prior works like SnapKV (Li et al., 2024) which are only limited to MHA.

## 4. Evaluation Methodology

**Models:** We evaluate MorphKV across four state-of-the-art LLMs chosen based on their complementary strengths:

- **Llama-3.1 8B Instruct** (Dubey et al., 2024): A model optimized for long-context tasks (128K token window) and coherent multi-turn dialogue.

- **Mistral-v0.2 7B Instruct** (Jiang et al., 2023): A

lightweight architecture designed for efficient deployment on consumer hardware.

- **Qwen2.5 7B Instruct** (Qwen et al., 2025): A model with multi-lingual English and Chinese proficiency.
- **Phi-4 14B** (Abdin et al., 2024): A model specialized in STEM reasoning through high-quality training data and curriculum learning.

This diversity ensures rigorous validation of MorphKV's robustness, scalability, and cross-architectural consistency.

**Setup:** We run experiments on an NVIDIA Grace Hopper node with an H200 GPU (96GB HBM3) and Grace CPU (116GB LPDDR5) interconnected via NVLink. We implement MorphKV using HuggingFace Transformers (Wolf, 2020) with FlashAttention-2 (Dao, 2023) for hardware-aware optimization, mirroring the configuration of prior KV cache works (Adnan et al., 2024; Li et al., 2024; Wang et al., 2024; Zhang et al., 2023).

**Benchmarks:**

- **Long-Response generation:** LongWriter (Bai et al., 2024) and LongGenBench (Liu et al., 2024a), which require synthesizing structured outputs (e.g., diaries, floorplans) based on input prompts.
- **Long-Context understanding:** LongBench (Bai et al., 2023), for tasks like code repository navigation and document summary with 16K-128K token contexts.

**Baseline:** We compare the performance and memory efficiency of MorphKV against SnapKV (Li et al., 2024), $H_2O$ (Zhang et al., 2023), and Full-Attention. SnapKV is the state-of-the-art for KV cache compression, while Full-Attention provides an upper bound on accuracy. However, SnapKV does not perform token eviction during the generation phase, making it less efficient for long-response tasks where cache management is critical. $H_2O$ applies KV cache pruning and is thus a more meaningful baseline for evaluating MorphKV in long-response settings. Prior works retain KV pairs across all attention heads, while MorphKV's compatibility with Grouped-Query Attention (GQA) enables a more memory-efficient approach as it only retains KV pairs across grouped-key heads, allowing us to assess trade-offs between KV cache size and retention of relevant tokens across different models and benchmarks.

**Implementation:** We implement MorphKV in Hugging-Face `transformers library` (Wolf, 2020), integrating it into the existing attention mechanism and leveraging FlashAttention (Dao, 2023) for efficient inference. To extract attention scores for older tokens, we compute partial attention weights for window queries within FlashAttention and store them as a lightweight KV cache extension. We update this cache during generation by appending new attention profiles and discarding the oldest ones.

# 5. Results

## 5.1. Long-Response: LongWriter Tasks

We evaluate MorphKV on open-ended, long-response text generation using the LongWriter (en) benchmark. Long-Writer covers tasks such as writing emails, blog posts, essays, and novels, with 60 prompts requesting responses ranging from 100 to 12000 words. For comparison, SnapKV retains 600 prompt tokens plus all decoded tokens across all attention heads, while $H_2O$ stores 600 decoded tokens per head. In contrast, MorphKV maintains a recent window of 30 tokens, a total KV cache capacity of 600 tokens, and supports two fusion strategies: *sum()* and *max()*.

### 5.1.1. PERFORMANCE

Performance is assessed using an LLM-based Judge (Mistral-Large-123B), with assigned scores across several criteria aggregated into a final metric. Table 1 shows that MorphKV outperforms $H_2O$, and SnapKV on Llama, Mistral, and Phi4, while achieving comparable performance for Qwen. MorphKV consistently excels over $H_2O$ in relevance (see Appendix A.4), demonstrating its ability to retain critical context, even under memory constraints.

| Model | Llama | Mistral | Phi4 | Qwen |
|---|---|---|---|---|
| $H_2O$ | 68.5 | 80.0 | 61.5 | 63.8 |
| SnapKV | 67.7 | 81.1 | 63.8 | **68.4** |
| MorphKV | **69.5** | **81.1** | **64.7** | 64.9 |
| Full-Attention | 66.5 | 81.3 | 62.9 | 66.2 |

*Table 1.* LongWriter: Comparison of LLM Judge Scores shows that MorphKV outperforms other methods by up to 4.5%, retaining important older tokens at a much smaller memory footprint.

### 5.1.2. KV CACHE SIZES

Figure 6 shows the normalized KV cache sizes relative to Full-Attention. On average, MorphKV reduces memory usage to $0.25\times$ that of Full-Attention, while $H_2O$ requires $1\times$ and SnapKV incurs significantly higher usage, up to $4\times$.

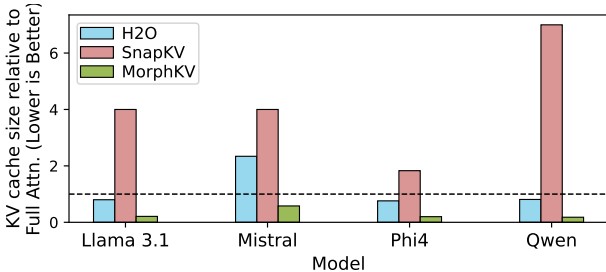

*Figure 6.* LongWriter: KV cache usage relative to Full-Attention. On average, MorphKV requires only $0.25\times$ the KV cache size, while $H_2O$ and SnapKV consume $1\times$ and $4\times$ respectively.

### 5.1.3. IMPACT OF INCREASING RESPONSE LENGTH

To evaluate the robustness of MorphKV for long responses, we compute the LLM Judge Score against increasing response lengths, as shown in Figure 7 for Mistral-7B. As the response length increases, performance declines across all methods due to the inherent challenges of generating extremely long text (Bai et al., 2024). However, MorphKV degrades more gradually: a $4\times$ increase in length reduces performance by $15\%$–$18\%$ for SnapKV and $H_2O$, whereas the performance only reduces by $10\%$ for MorphKV. Notably, MorphKV maintains a constant KV cache size regardless of the response length, contributing to its efficiency and robustness over extended text generations.

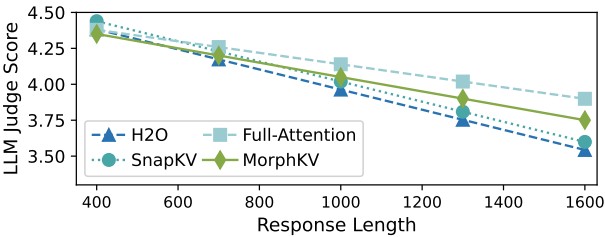

*Figure 7.* LongWriter: LLM Judge Score versus response lengths. MorphKV is more robust against increasing response lengths compared to SnapKV or $H_2O$ as it uses an adaptive KV selection algorithm, discarding those KVs which don't contribute significantly to the contextual flow. Notably, MorphKV maintains a fixed KV cache size regardless of response length.

### 5.2. Long-Response: LongGenBench Tasks

LongGenBench contains structured long-response tasks for temporal and spatial categories. The temporal category is divided into Diary Entry and Menu planning tasks, while the spatial category includes Skyscraper Design and Urban Planning. The original dataset has 400 samples (100 from each sub-category). For a fair comparison under limited resources, we select 40 samples, with ten from each sub-category, and use greedy decoding capped at 8K tokens. SnapKV employs all attention heads with a 32-token window and a total KV cache capacity of 1K tokens per head, whereas $H_2O$ keeps 4K tokens in the cache, also maintaining all attention heads. In contrast, MorphKV uses a 200-token recent window and a 4K-token total capacity across GQA heads, adopting *max()* fusion due to its consistently higher performance than *sum()*. The larger window enables MorphKV to retain critical distant tokens during generation (refer to Appendix A.2 for further information).

### 5.2.1. PERFORMANCE

Table 2 shows the performance of MorphKV over prior works. LongGenBench uses a rigorous evaluation suite

| | Model | CR (%) | Accuracy (%) | | | |
|---|---|---|---|---|---|---|
| | | | Once | Range | Periodic | Avg. |
| Llama | $H_2O$ | 64 | 45 | 60 | **27** | 44 |
| | SnapKV | 64 | 50 | 55 | 26 | 44 |
| | MorphKV | **64** | **50** | **61** | 24 | **45** |
| Mistral | $H_2O$ | 71.2 | 57 | 60 | 32 | 50 |
| | SnapKV | 71 | 55 | 57 | 36 | 49 |
| | MorphKV | **71.2** | 57 | **62** | **36** | **52** |
| Qwen | $H_2O$ | **55** | **46** | 51 | 28 | 42 |
| | SnapKV | 53 | 44 | 46 | 28 | 39 |
| | MorphKV | 51 | 43 | **68** | **30** | **47** |

*Table 2.* LongGenBench: Performance comparison of MorphKV, against SnapKV, $H_2O$. MorphKV achieves better scores across all evaluation metrics i.e., Completion Rate (CR), and all grouped accuracy metrics (Accuracy Once, Range, Periodic, and Average).

to assess response quality. This includes information recall about singular instances (Accuracy Once), range of instances (Accuracy Range), periodic instances (Accuracy Periodic), and their average (Average Accuracy), while Completion Rate (CR) quantifies the percentage of tasks successfully completed. MorphKV generally outperforms or matches SnapKV, and $H_2O$ on all models and metrics. Notably, SnapKV retains all prompt tokens due to its ample cache budget. It also keeps track of every decoded token, effectively replicating Full-Attention for these tasks.

### 5.2.2. KV CACHE SIZES

Figure 8 shows the KV cache sizes for $H_2O$, SnapKV, and MorphKV relative to Full-Attention. On average, MorphKV achieves significant memory savings, requiring only $0.55\times$ the memory usage with Full-Attention, while $H_2O$ and SnapKV require $1.22\times$, and upto $5.01\times$ the cache size of Full-Attention, respectively.

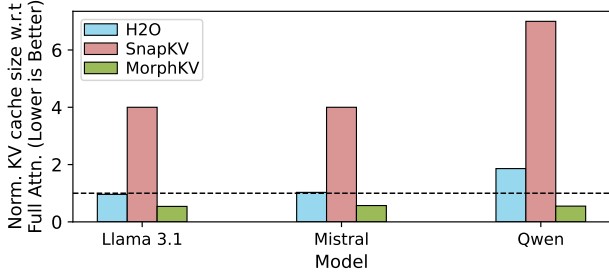

*Figure 8.* LongGenBench: KV cache usage relative to Full-Attention. SnapKV incurs up to $13\times$ higher due to the extensive retention of KV pairs, while MorphKV maintains a constant cache.

|  | Model | 2wmqa | drdr | hpqa | mnews | mfqaen | mfqazh | musq | nqa | pcnt | prt | qsp | qms | sams | tqa | vcs |
|---|---|---|---|---|---|---|---|---|---|---|---|---|---|---|---|---|
| Llama | SnapKV | **16.0** | 22.0 | 14.9 | 25.6 | 25.4 | 18.7 | **10.7** | **32.2** | **7.6** | **98.4** | 11.7 | 23.1 | **42.9** | **91.7** | 14.2 |
| Llama | MorphKV | 14.9 | **22.5** | **15.9** | **26.6** | 25.7 | **19.9** | **10.7** | 31.9 | 7.5 | 97.8 | **11.9** | **23.6** | **42.9** | 91.5 | **15.2** |
| Llama | Full-Attention | 16.5 | 30.0 | 16.7 | 26.8 | 27.4 | 20.1 | 11.4 | 32.0 | 6.9 | 97.7 | 13.2 | 23.6 | 43.7 | 91.6 | 16.1 |
| Mistral | SnapKV | 26.6 | 23.7 | 40.5 | 26.0 | **48.8** | 41.3 | **18.3** | 25.6 | 2.5 | **88.6** | **31.0** | **23.8** | 41.9 | **86.3** | 13.5 |
| Mistral | MorphKV | **26.7** | **23.9** | **40.8** | **26.6** | 48.4 | **43.0** | 16.7 | **26.7** | **3.0** | 85.9 | 30.9 | 23.6 | **42.3** | **86.3** | **13.7** |
| Mistral | Full-Attention | 27.1 | 30.4 | 43.0 | 27.1 | 49.2 | 48.3 | 18.8 | 26.7 | 2.8 | 87.0 | 33.0 | 24.2 | 42.8 | 86.2 | 15.2 |
| Phi4 | SnapKV | 22.3 | **24.2** | **19.5** | 25.0 | 38.0 | **47.2** | 5.2 | 20.5 | **12.6** | 63.9 | **32.4** | 22.1 | 47.2 | 90.5 | 11.4 |
| Phi4 | MorphKV | **22.6** | 24.1 | 19.3 | **25.5** | **38.2** | 46.4 | **6.2** | **21.0** | **12.6** | **64.3** | 31.2 | **22.4** | **47.6** | **90.6** | **12.3** |
| Phi4 | Full-Attention | 22.2 | 29.0 | 19.6 | 25.9 | 38.2 | 48.9 | 6.0 | 20.7 | 11.6 | 63.3 | 33.3 | 22.9 | 48.2 | 90.4 | 13.4 |

*Table 3.* LongBench: Performance comparison of MorphKV, SnapKV, and full attention across different models. MorphKV achieves higher accuracy in most micro-benchmarks, as its KV selection algorithm minimizes redundancy and noise in the attention profile.

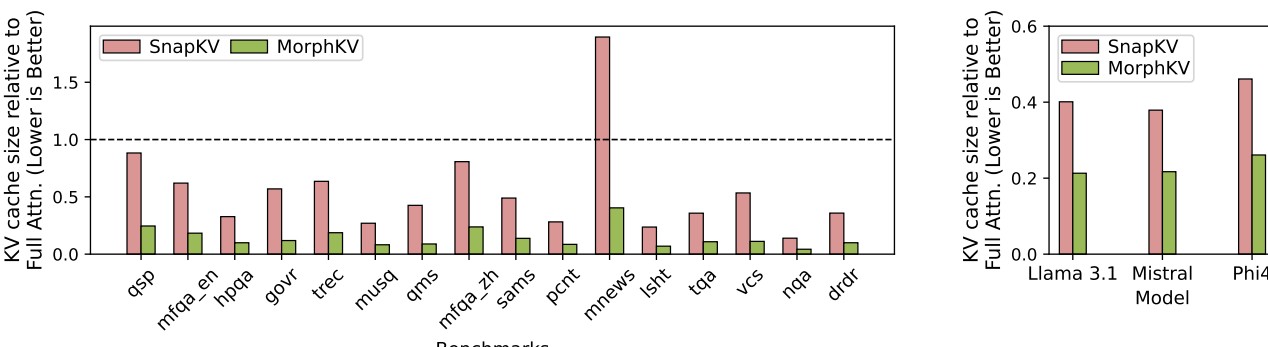

*Figure 9.* LongBench: (a) Llama3.1-8B Instruct KV cache sizes of SnapKV, and MorphKV relative to full attention. On an average, SnapKV has a KV cache size of $0.42\times$, whereas MorphKV is $0.15\times$ compared to Full-Attention (b) Average KV cache sizes of SnapKV and MorphKV relative to full attention across different models. MorphKV yields comparable performance to SnapKV at roughly 50% lower KV cache budget in a long-context setting, where the prompt is significantly larger than the response.

## 5.3. Long-Context: LongBench Tasks

Besides being memory-efficient for long-response tasks, MorphKV also offers competitive performance as the state-of-the-art prompt KV compression for long-context tasks. We evaluate MorphKV, SnapKV, and Full-Attention across benchmarks in LongBench. LongBench is a comprehensive, bilingual, multitask benchmark suite used to evaluate LLMs for processing extended contexts. It comprises datasets across six task categories in English and Chinese, with an average prompt length of nearly 6K tokens. For MorphKV, we set the recent window to 32 tokens and fix its total cache capacity at 2K tokens, and use the *sum()* fusion. In contrast, SnapKV preserves 1024 tokens from the prefill phase and all decoded tokens across all attention heads.

### 5.3.1. Performance

Table 3 shows that MorphKV generally matches or outperforms SnapKV across most datasets. Notably, MorphKV consistently surpasses SnapKV for MultiNews on all models.

Moreover, for tasks like Phi4-2WikiMQA, Phi4-Passage-Count, and Phi4-TriviaQA, MorphKV even exceeds Full-Attention performance while using only 20% of the memory capacity. This suggests that larger models (e.g., Phi4 with 14B parameters) can better leverage the dynamic token selection in MorphKV to capture essential information.

### 5.3.2. KV CACHE SIZES

Figure 9 compares the average KV cache memory usage of SnapKV and MorphKV relative to Full-Attention across all LongBench datasets. MorphKV achieves up to $2\times$ memory savings over SnapKV and up to $5\times$ over Full-Attention. Notably, for datasets like MultiNews, SnapKV requires $2\times$ more memory than Full-Attention because it retains KV pairs across all heads, whereas MorphKV operates at just $0.4\times$ the memory of Full-Attention, benefiting from dynamic eviction and GQA compatibility. Designed for GQA, MorphKV supports $2\times$ more tokens while using only half the KV cache capacity of SnapKV.

## 6. Runtime and System-Level Trade-offs

MorphKV incurs runtime overhead during response-generation as it introduces additional computation steps that depend on the attention profile of the recent window tokens, necessitating additional memory accesses. To minimize this overhead, the implementation of MorphKV employs dedicated CUDA streams to prefetch the attention profile ahead of time. By default, MorphKV performs token-eviction at each generation step to precisely manage KV cache memory usage, crucial for long-context and long-response tasks.

Table 4 shows that the memory savings with MorphKV significantly outweigh the runtime overhead, improving the overall system throughput. Furthermore, this runtime cost can be amortized by evicting tokens less frequently (see Appendix A.5), albeit at higher memory usage. These trade-offs highlight that KV cache compression techniques must navigate several system-level considerations, and optimizing for a single metric alone is inadequate for practical adoption.

| Metric | SnapKV | $H_2O$ | MorphKV |
|---|---|---|---|
| Runtime (lower is better) | **1×** | 1.62× | 1.50× |
| Memory (lower is better) | 1× | 0.26× | **0.14×** |
| Accuracy (higher is better) | 1× | 0.94× | **1.01×** |
| Throughput (higher is better) | 1× | 2.40× | **4.68×** |

*Table 4.* LongBench: Despite introducing runtime overhead, MorphKV achieves substantial memory savings, leading to higher overall system throughput, while maintaining response accuracy

## 7. Discussion

### 7.1. Variation with Design Parameters

The design of MorphKV consists of three key design parameters, namely, fusion function used to create the attention profile, recent window size, and total KV cache budget. These hyperparameters collectively influence the overall performance. MorphKV shows minimal sensitivity to variations in its hyperparameters, highlighting its robustness across various configurations. We refer the reader to Appendix A.1, A.2, and A.3 for detailed discussions on the sensitivity of MorphKV's performance to the choice of fusion function, window sizes, and KV cache budget respectively.

### 7.2. Outperforming Full Attention

KV cache compression methods, despite relying on a subset of context tokens, often outperform full attention in practice (Li et al., 2024; Cai et al., 2024). This stems from their ability to focus on the most relevant tokens, reducing the influence of less useful ones. This is especially beneficial for long-response scenarios, where noise due to irrelevant tokens accumulates over time, leading to a noticeable decline

in output quality, or *degeneration* (Holtzman et al., 2019). We examine the N-gram repetition rate in responses from the Llama3.1-8B Instruct model to measure this degeneration in generated responses. A higher repetition rate indicates higher degradation in output quality. Table 5 shows that MorphKV has the lowest repetition rate as it continues to prune the KV cache during token generation. In contrast, SnapKV and full attention exhibit greater redundancy as they retain many more tokens in the KV cache.

| Metric | MorphKV | SnapKV | Full-Attention |
|---|---|---|---|
| Repetition Rate (N=10) | **68%** | 89% | 89% |

*Table 5.* LongWriter: Degeneration in Llama3.1-8B responses measured using N-gram (N=10) repetition. MorphKV reduces repetition by dynamically evicting tokens, limiting contextual noise.

## 8. Future Work

In addition to token selection, prior work has shown that KV cache can further be optimized across attention layers (Cai et al., 2024), and attention heads (Fu et al., 2024; Feng et al., 2024). MorphKV is complementary to these works, and integrating them would yield further memory savings. While a detailed evaluation of integrating MorphKV across other optimization axes is deferred to future work, we present a preliminary analysis with layer-aware considerations in Appendix A.6. Furthermore, to minimize the runtime cost with MorphKV, we plan to integrate MorphKV within the Flash Attention kernel, improving inference efficiency while preserving accuracy and memory benefits.

## 9. Conclusion

The growing memory footprint of KV caches in LLMs poses a critical bottleneck for long-context and long-response tasks. In this paper, we propose MorphKV that addresses this challenge by introducing a dynamic, correlation-aware token selection mechanism that maintains a constant-sized KV cache while preserving contextual coherence. MorphKV leverages attention profiles of recent tokens to retain only the most relevant distant tokens. Our studies on long-response tasks show 52.9% memory savings and 18.2% higher accuracy on average compared to state-of-the-art prior works. Our experiments demonstrate that MorphKV scales efficiently with response length, degrading only 10% in performance even as outputs grow to 12K tokens, compared to 15–18% degradation for state-of-the-art prior works. Furthermore, MorphKV's compatibility with GQA enables 4× greater memory efficiency than MHA-based approaches, making it practical for real-world deployment. These advances position MorphKV as a practical inference-time solution, balancing accuracy and memory usage without sacrificing the ability to capture long-range dependencies.

## Acknowledgments

This research has been supported by computing support on the Vista GPU Cluster through the Center for Generative AI (CGAI) and the Texas Advanced Computing Center (TACC) at the University of Texas at Austin. We thank the generous support from the Cockrell School of Engineering and the AMD endowment at the University of Texas at Austin. This work is supported in part by the Cisco Research Award. Prashant J. Nair is supported by Intel Transformation Server Architecture (TSA) and the Natural Sciences and Engineering Research Council of Canada (NSERC) [funding reference number RGPIN-2019-05059] Grants. We acknowledge Won Joon Yun for editorial feedback and Sourish Wawdhane for the code structuring effort.

## Impact Statement

This work aims to advance the field of Machine Learning by reducing the memory footprint of large language model (LLM) inference while maintaining accuracy. By lowering hardware constraints, our technique potentially broadens the applicability of LLMs to more users and domains, including creative writing, data-intensive research, and education. However, it also amplifies existing concerns: easier access to powerful LLMs may exacerbate issues such as misuse of misinformation, large-scale automation of personalized or sensitive communication, and environmental impact due to increased computational usage. While the proposed method does not introduce new ethical risks beyond those already associated with LLMs, we encourage developers and practitioners to deploy it responsibly, carefully considering the contexts in which LLMs are employed and the societal implications of further democratizing access.

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

# A. Appendix

## A.1. Impact of Fusion Function: *sum()* Versus *max()*

MorphKV considers two fusion functions for deriving the final attention profile, namely, $F_i$: $sum()$ and $max()$. In this subsection, we discuss their impact on the performance.

### A.1.1. ON LONG-CONTEXT TASKS

We compare both $sum()$ and $max()$ fusion using the LongBench suite on the Llama3.1-8B model, with a recent window configuration of 32 tokens and KV cache capacity of 1K tokens. Table 6 shows per-dataset performance. On average, $max()$ fusion outperforms $sum()$ by about 1%, and up to 2.7% on QMSum. Datasets such as 2WikiMQA, MultiNews, Passage Count, Passage Retrieval (En), QMSum, and TriviaQA often demand sharply focused retrieval or reasoning. A single strongly attended token in these tasks can suffice to link crucial context, making $max()$ advantageous. In contrast, $sum()$ tends to retain past tokens which are preferred by majority of the window tokens. Consequently, $max()$ better captures a small set of pivotal tokens spread over a large distance (longer sequence of tokens).

| Llama3.1 | 2wmqa | drdr | hpqa | lsht | mnews | mfqaen | mfqazh | musq | nqa | pcnt | prt | qsp | qms | sams | trec | tqa | vcs |
|---|---|---|---|---|---|---|---|---|---|---|---|---|---|---|---|---|---|
| max_fused | **14.9** | 21.0 | 14.9 | 33.5 | **25.6** | 24.2 | 17.9 | 9.3 | 32.0 | **8.0** | **97.6** | 10.0 | **23.5** | 42.1 | 46 | **91.8** | 14.5 |
| sum_fused | 14.6 | **22.0** | **15.0** | **35.5** | **25.6** | **25.4** | **19.2** | **9.9** | **32.2** | 7.9 | 96.9 | **10.5** | 22.9 | **43.1** | **49** | **91.8** | **14.8** |

*Table 6.* LongBench: Llama3.1-8B-Instruct comparison of MorphKV under different fusion functions with the same cache budget.

### A.1.2. ON LONG-RESPONSE TASKS

We similarly compare the sensitivity of $sum()$ and $max()$ for LongWriter tasks which contain essay-style long response prompts. For our studies, we fix the recent window to be 30 tokens, and KV cache capacity to a total of 600 tokens. As shown in Table 7, $sum()$ fusion tends to be more effective for most models, except Qwen2.5 where $max()$ excels in certain metrics. LongWriter tasks are typically open-ended, causing $max()$ to emphasize specific tokens that are not always universally relevant. Conversely, $sum()$ aggregates attention across the recent window, providing a broader (though slightly noisier) context that suits open-ended generation.

| | Fusion | Relevance | Accuracy | Coherence | Clarity |
|---|---|---|---|---|---|
| Llama | max | 83.8 | 81.3 | 57.1 | 64.2 |
| Llama | sum | **89.2** | **81.7** | **63.3** | **71.3** |
| Mistral | max | 91.7 | 86.7 | 82.9 | 82.1 |
| Mistral | sum | **92.5** | **89.2** | **84.2** | **85.4** |
| Phi4 | max | **62.9** | 79.6 | 68.3 | 72.1 |
| Phi4 | sum | 62.5 | **80** | **70.4** | **75.0** |
| Qwen | max | 83.3 | **70.8** | 58.3 | **60.4** |
| Qwen | sum | **85.4** | 70.4 | **58.3** | 59.1 |

*Table 7.* LongWriter: Sensitivity to $sum()$ versus $max()$ fusion across different models. These fusion functions dictate how attention scores are aggregated to make the final attention profile.

## A.2. Impact of Window Size on Long-Response Tasks

We evaluate the impact of window size on MorphKV's performance. Intuitively, recent-window tokens determine which tokens to retain from the distant past. Thus, a larger window allows capturing of more diverse information from the past.

Particularly, for LongGenBench, this effect is evident since the prompts contain lot of information which might be needed at a much later point in the generation. Hence, we run the LongGenBench suite for Llama3.1-8B, Mistral-7B and Qwen2.5

models for two variants of MorphKV, both using max fusion and a total capacity of 4K tokens. The first configuration uses a 32-token window, while the second uses a 200-token window.

As shown in Table 8, changing the window size significantly impacts evaluation metrics for all Llama, Mistral, and Qwen models. This is due to the fact that a very small window does not suffice for capturing extensive amounts of distant information present in a typical LongGenBench prompt (such as specific details about different floors in a building, specific Menu items etc.), because very small windows tend to capture local context, while failing to capture more distant context. Therefore, a larger window is more effective since it allows the model to retain diverse pieces of information even at very large response sizes. For instance, a window of size 200 lets Llama3.1 recall accurate instructions regarding the 96th floor of a building, in spite of already generating extensive descriptions of the previous 95 floors. On the other hand, with a window size of 32 tokens, the model struggles to maintain consistency with the input request, and generates generic responses after certain number of floors, thereby losing on accuracy.

Note that the Completion Rates for both window sizes are comparable. This is because the Completion Rate measures the number of times the model was able to generate what it was expected to (for example, how many floors did the model generate the floor plan for out of the requested 100 floors). This is a relatively simpler task, and a smaller window can keep track of such information (for instance, by simply retaining the floor number of the last floor it generated the plan for). Consequently, we do not observe substantial differences in the Completion Rate metric.

| | Config | CR(%) | Accuracy(%) | | | |
| | | | Once | Range | Periodic | Avg. |
|---|---|---|---|---|---|---|
| Llama | (32, 4K) | 64 | 43 | 56 | **27** | 42 |
| | (200, 4K) | **64** | **50** | **61** | 24 | **45** |
| Mistral | (32, 4K) | 71 | 57 | 60 | 32 | 50 |
| | (200, 4K) | **71** | **57** | **62** | **36** | **52** |
| Qwen | (32, 4K) | **52** | 41 | 43 | **34** | 40 |
| | (200, 4K) | 51 | **43** | **68** | 30 | **47** |

*Table 8.* LongGenBench: Sensitivity of evaluation metrics with window size across different models (CR: Completion Rate). A small window is insufficient for capturing large-amounts of distant information in long-response tasks, leading to a degradation in model accuracy for all models. However, Completion rate remains comparable across window sizes as it only measures the model's ability to complete the response as expected, without considering the relevance or correctness of the generated output.

### A.3. Robustness Against KV Cache Compression

To assess the impact of KV cache compression on MorphKV versus SnapKV, we run ablation studies on a subset of benchmarks within the LongBench suite. We record the resulting performance across Llama3.1-8B and Mistral-7B models. For both MorphKV and SnapKV, the KV cache budget is varied from 1% to 7% with respect to full attention KV cache size. MorphKV uses a window of 32 tokens, with $sum()$ as the fusion function. SnapKV also uses the same window size of 32 tokens, but maintains KV cache across all attention heads. This enables MorphKV to store $4\times$ as many tokens as SnapKV at the same cache capacity.

Figure 10 shows the mean and individual benchmark scores for both models under varying compression scenarios. At very low KV cache budgets, we observe a drop of more than $50\%$ on average between MorphKV and SnapKV, for benchmarks like NarrativeQA, this difference reaches upto $88\%$. This disparity indicates that MorphKV is significantly more effective at retaining crucial context information compared to SnapKV under tight memory constraints. Even with larger budgets, MorphKV consistently outperforms SnapKV, demonstrating the robustness and reliability of its design.

The input prompt for long-response tasks is typically very small, and SnapKV does not evict KV cache during decoding, hence we exclude a similar analysis for these tasks.

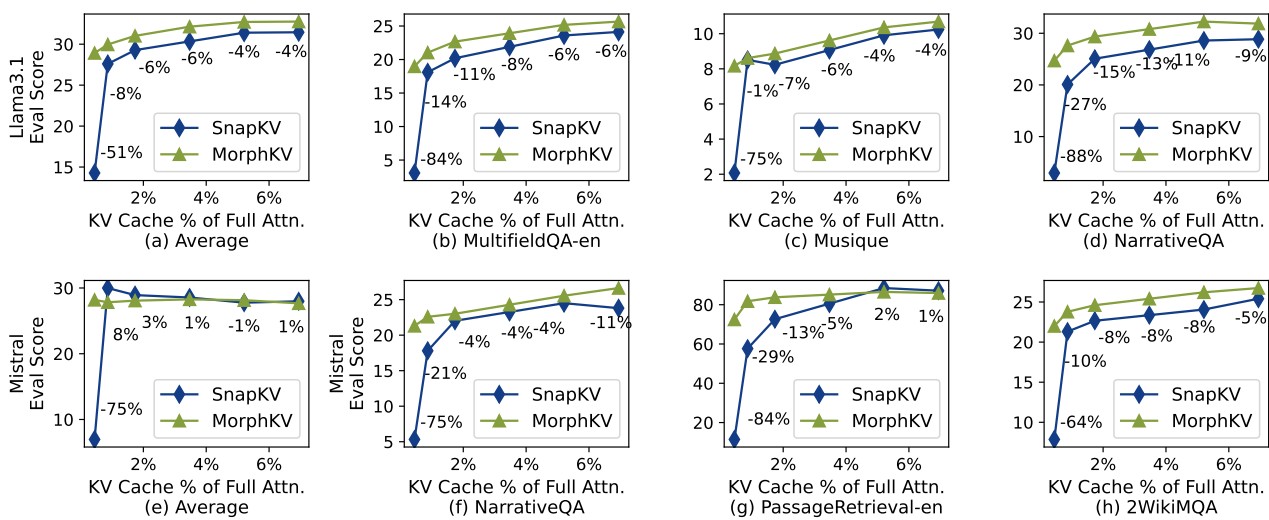

*Figure 10.* Comparison of MorphKV versus SnapKV for Llama3.1-8B, and Mistralv0.2-7B. (a)-(d) show results for Llama3.1-8B on Average, MultifieldQA-en, Musique, and NarrativeQA respectively, while (e)-(h) show results for Mistralv0.2-7B on Average, NarrativeQA, PassageRetrieval-en, and 2wikimqa respectively. MorphKV consistently beats SnapKV across varying KV cache budgets.

## A.4. LongWriter: LLM Judge Scores for Different Models and Metrics

Table 9 provides detailed LLM Judge Scores across various criteria. All configurations use a KV cache capacity of 600 tokens, with a recent window size of 30 tokens for both SnapKV and MorphKV. MorphKV consistently outperforms $H_2O$ on *Relevance*, highlighting its effectiveness in retaining imporarant tokens. Qwen2.5 differs from other models as it employs GQA with $7\times$ less heads than the default MHA configuration, potentially introducing partial information loss when MorphKV operates under fewer heads, explaining why MorphKV performs poorer compared to SnapKV for this model.

| | Model | Relevance | Accuracy | Coherence | Clarity | Breadth and Depth | Reading Experience | Total |
|---|---|---|---|---|---|---|---|---|
| Llama3.1-8B | $H_2O$ | 84.6 | **81.7** | **63.3** | 71.7 | 54.2 | **55.4** | 68.5 |
| | SnapKV | 85.8 | 80.4 | **63.3** | **73.3** | 48.8 | 54.6 | 67.7 |
| | MorphKV | **89.2** | **81.7** | **63.3** | 71.2 | **57.1** | 54.6 | **69.5** |
| | Full-Attention | 86.2 | **81.7** | 57.9 | 71.2 | 49.2 | 52.5 | 66.5 |
| Mistral-7B | $H_2O$ | 91.7 | **89.6** | 81.2 | 83.3 | 60.4 | 73.8 | 80.0 |
| | SnapKV | 90.4 | **89.6** | 84.6 | 84.2 | **61.7** | **76.2** | 81.1 |
| | MorphKV | 92.5 | 89.2 | 84.2 | **85.4** | 60.4 | 75.0 | 81.1 |
| | Full-Attention | **93.8** | 88.8 | **85.0** | 84.6 | 60.4 | 75.0 | **81.2** |
| Phi4 | $H_2O$ | 59.2 | 77.9 | 63.8 | 70.8 | 40.4 | 57.1 | 61.5 |
| | SnapKV | **66.7** | 78.3 | 68.3 | 71.2 | 42.5 | 55.8 | 63.8 |
| | MorphKV | 62.5 | **80** | **70.42** | **75** | 41.2 | **58.75** | **64.65** |
| | Full-Attention | 65.0 | 78.3 | 63.8 | 72.9 | **43.8** | 53.8 | 62.9 |
| Qwen2.5 | $H_2O$ | 85.0 | 67.1 | 54.6 | 56.2 | 67.1 | 52.5 | 63.8 |
| | SnapKV | **87.7** | **72.5** | **60.6** | **63.6** | **68.2** | **57.6** | **68.4** |
| | MorphKV | 85.4 | 70.4 | 58.3 | 59.2 | 63.8 | 52.5 | 64.9 |
| | Full-Attention | 86.4 | 69.5 | 59.8 | 61.0 | 64.4 | 56.4 | 66.2 |

*Table 9.* LongWriter: LLM Judge Scores by model across multiple metrics.

## A.5. Coarse-Grained Token-Eviction

MorphKV performs a dynamic token eviction at every generation step to maintain a constant-sized KV cache. However, this fine-grained approach incurs runtime overhead as the algorithm must run at every timestep, necessitating memory accesses

to load the attention profile. While MorphKV reduces this overhead by employing dedicated CUDA streams to prefetch the attention profile, further reductions could be achieved by invoking the algorithm at certain time intervals. Table 10 presents the performance impact when the token-selection policy is applied at every 8th generation-step (eg., 8, 16, 32 etc.). Adopting a coarser token-eviction results in runtime improvement, albeit at reduced accuracy. This trade-off suggests that the balance between latency, and accuracy can be dynamically tuned to suit task-specific requirements at runtime.

| MorphKV | Completion Rate | Avg. Accuracy | Runtime |
|---|---|---|---|
| default | 72.1% | 47.0% | 1× |
| coarse-grained eviction (step=8) | 71.9% | 46.1% | 0.82× |

*Table 10.* LongGenBench: Performance of MorphKV with different token eviction policies on Mistral-7B. Coarse-grained selection incurs a slight degradation in long-response generation but yields better runtime compared to default (per-timestep) eviction.

## A.6. Selective Layer Retention with MorphKV

MorphKV only preserves those distant tokens which are heavily attended by the recent window tokens. However, KV cache can also be optimized across attention layers. Prior works (Cai et al., 2024; Wang et al., 2024) have shown that early layers capture more critical information than later ones. Building on this insight, we perform a preliminary experiment where we disable MorphKV for the first three layers i.e., the first three layers are retained for all tokens, to emulate a layer-sensitive strategy. Table 11 shows that this configuration outperforms the default MorphKV configuration, although with an additional 10% memory footprint.

While integrating MorphKV with orthogonal techniques which optimize the KV cache across attention layers (Cai et al., 2024; Wang et al., 2024), and attention heads (Fu et al., 2024; Feng et al., 2024) can further improve performance, such integration necessitates a comprehensive understanding of their trade-offs, and careful parameter tuning. We defer this exploration to future work.

| MorphKV) | Completion Rate | Avg. Accuracy |
|---|---|---|
| default | 70.51% | 44.2% |
| layer-aware optimization | 71.96% | 46.1% |

*Table 11.* LongGenBench: Performance of MorphKV improves with selection (first three layers) layer retention on Mistral-7B, although at an additional 10% memory overhead. Default configuration of MorphKV has a capacity of 2000 tokens.

