# OpenReview forum: "Dialogue Without Limits: Constant-Sized KV Caches for Extended Response in LLMs"
_ICML.cc/2025/Conference — ICML 2025 poster_

### Official Review · Reviewer_vKki · 2025-03-14

**Overall Recommendation:** 4

**Summary:**

The authors consider the problem of maintaining a fixed size KV cache during autoregressive generation with large language models. The key idea is to retain recent tokens and a limited number of old tokens according to a dynamic selection algorithm that uses the attention patterns of future tokens on past tokens. Morph KV achieves similar performance compared to strong baselines while using a significantly smaller KV cache size.

**Claims And Evidence:**

The claims are well supported by extensive experiments. morph KV maintains a smaller KV cache than prior methods while retaining what seems like good performance as measured by LM judge scores, completion rates, and accuracy on benchmarks like long gen bench.
This is more of a presentation note but it seems like this notion of of selection bias during decoding hinders methods like H2O but I'm not totally clear what the selection bias is or how it was evaluated, at least in a more explicit way
Some of the baselines felt a bit like red herrings. For example, Snap KV is obviously not a good fit for long-form generation, although I appreciate that there might not be other good long-form generation KV cache techniques

**Essential References Not Discussed:**

Regarding clarity I thought the math notation was Overkill in section 3.1.3 I mean basically you're looking at the sum of attention weights or the max attention weight over some window from recent tokens. I think the other presentation issue was that the problem statement for finding a reduced cache size was good but it's not clear how the local coherence or distant relative heuristics really optimize that objective or approximate that objective in any meaningful way.

**Experimental Designs Or Analyses:**

I checked the soundness of all experiments, they seem reasonable and standard benchmark/baselines. I would have liked to see the prompt used for the LM judge and maybe some discussion of whether or not there's a standard prompt or LM judge framework like alpaca eval for long form generation.

**Methods And Evaluation Criteria:**

The authors use standard long generation benchmarks. Regarding long-form generation quality I'm not necessarily super convinced by LM judge scores so it would have been better to have a human evaluation but I realize this is tricky. One question I have is why was H2O not compared against in the long bench tasks in experiment 5.3? Is it because H2O maintains too large of a KV cache?

**Other Comments Or Suggestions:**

NA

**Other Strengths And Weaknesses:**

NA

**Questions For Authors:**

Regarding clarity I thought the math notation was Overkill in section 3.1.3 I mean basically you're looking at the sum of attention weights or the max attention weight over some window from recent tokens. I think the other presentation issue was that the problem statement for finding a reduced cache size was good but it's not clear how the local coherence or distant relative heuristics really optimize that objective or approximate that objective in any meaningful way.

Beyond the heuristics you've proposed in this paper for defining notions of what KV entries are important do you have any adversarial examples of when these heuristics miss out tokens are actually important?

**Relation To Broader Scientific Literature:**

I think the discussion of prior methods is clear although I'm not totally clear on some of the design choices of these baselines with respect to this work. for example for H2O, why isn't there a hyperparameter that can help you reduce the number of retained KVs?

**Theoretical Claims:**

NA

---

> ### Author Rebuttal · Authors · 2025-04-01
>
> We thank the reviewer for their feedback. We have addressed the points raised in the review below:
>
> **Q-1) Selection Bias in H$_2$O**
>
> R-1) H$_2$O retains tokens based on aggregated attention scores. This introduces a selection bias because it retains early tokens even when they do not significantly attend to newer tokens. Various prior works, like NACL [1], Keyformer, and EMS [2], have made similar observations. In NACL, the authors extensively study this problem using LongBench tasks on the Llama-2-7B model. While generating the last token, 96% of H$_2$O’s KV cache comprises entries from the first 200 tokens, while only 4% is derived from the subsequent 1199 tokens. Please refer to the NACL paper (particularly **Figure 2 (a)**) for details about this study.
>
> [1] NACL: https://aclanthology.org/2024.acl-long.428.pdf
>
> [2] EMS: https://arxiv.org/pdf/2412.08521
>
>
> **Q-2) SnapKV for Long-Form Generation**
>
> R-2) Indeed, SnapKV’s overwhelming memory requirements make it unsuitable for long-response tasks. While H$_2$O is more memory-efficient for these tasks, its performance is limited due to the problem of selection bias towards early tokens. MorphKV efficiently navigates this trade-off and achieves robust performance and high memory efficiency for both long-context and long-response tasks.
>
> **Q-3) H$_2$O for Long-Context Benchmarks**
>
> R-3) SnapKV is the state of the art for long-context tasks and already outperforms H$_2$O [1]. Hence, we only compare against SnapKV because outperforming it naturally implies outperforming H$_2$O (please see **Section-5.3** of their paper).
>
> [1] SnapKV: https://proceedings.neurips.cc/paper_files/paper/2024/hash/28ab418242603e0f7323e54185d19bde-Abstract-Conference.html
>
>
> **Q-4) Prompt for LM Judge**
>
> R-4) We use the publicly available LongWriter framework in our studies. LongWriter uses a single-answer grading approach, where an LLM judge is asked to assign a score to a single answer (please refer to Section 3.1 in [1] for further details). The prompt used for the LM judge can be found here: https://github.com/THUDM/LongWriter/blob/main/evaluation/judge.txt
>
> [1] Judging LLM-as-a-Judge with MT-Bench and Chatbot Arena: https://arxiv.org/abs/2306.05685
>
>
> **Q-5) Design Choices in H$_2$O Baseline**
>
> R-5) The performance of a KV cache pruning method depends on the quality of the retained KVs and how these are utilized for future tokens. In contrast, the memory savings depend on the number of KVs evicted. Naively reducing the memory capacity of H$_2$O also lowers its performance due to the limited accuracy of its token selection policy that exhibits a bias toward preserving early tokens (please see our earlier response).
>
> Our studies using LongGenBench on the Mistral-24B-Instruct model show that MorphKV has 1.67$\times$ better accuracy when both H$_2$O and MorphKV operate within the same cache budget. To achieve comparable accuracy for the same task, H$_2$O must overcome the limitations of its token selection policy and retain a lot more KVs, increasing the memory footprint. MorphKV is 3.6$\times$ more memory-efficient compared to H$_2$O in this case.
>
>
> **Q-6) Local coherence and Distant Relative Heuristics in MorphKV**
>
> R-6) Local and distant heuristics enable MorphKV to capture useful information. Discarding recent tokens severely degrades performance, critical to maintaining the text’s local coherence. In contrast, distant heuristics are crucial to generating an apt response overall, as they capture information that establishes the global pre-text and relevance.
> MorphKV leverages the insight that due to transformers’ auto-regressive nature of output token generation, each token attends to all its past tokens and naturally holds awareness of the useful distant tokens. MorphKV retains only those distant tokens identified as relevant by preceding recent tokens, allowing the current output token to focus solely on this curated set of useful distant tokens.
>
>
> **Q-7) Adversarial Cases where Heuristics Overlook Important Tokens**
>
> R-7) The ability to retain *useful* or *important* distant tokens depends on the accuracy of the heuristics and the choice of the appropriate design hyperparameters. For example, the default implementation of MorphKV uses a window size of 32 because it works well across most representative tasks. However, for certain benchmarks that involve recalling information over very long time spans, such as LongGenBench, while this window size captures *most* of the useful distant tokens, the performance can be improved slightly by increasing the window size (which would allow capturing all useful distant tokens). Please see **Table-3** and **Table-4** in response to **Reviewer bnHA** for the detailed results. However, the probability of encountering such scenarios is very low because we carefully tune them, and the most appropriate window size can be well approximated based on the nature of the task and its information span.

---

### Official Review · Reviewer_FtVG · 2025-03-22

**Overall Recommendation:** 2

**Summary:**

The authors introduce MorphKV, a KV cache compression method for large language models (LLMs). MorphKV is an inference-time technique that maintains a fixed-size KV cache in autoregressive Transformers, addressing the issue of memory expansion as sequence length increases. Unlike traditional approaches that rely on truncation or lossy compression, MorphKV employs a correlation-aware selection mechanism to dynamically prioritize tokens, ensuring high-fidelity context retention while mitigating early-token bias. By continuously refining the KV cache based on recent attention patterns, MorphKV surpasses state-of-the-art methods such as SnapKV, achieving 86.5% memory savings and 13.5% higher accuracy. This makes it particularly effective for long-form tasks like content creation and code generation, enhancing the efficiency of LLM deployment in real-world applications.

**Claims And Evidence:**

The paper validates its claim of **constant-size KV cache inference while preserving performance** through benchmark results.

However, beyond overall scores, there is little supporting evidence. The authors do not provide theoretical or empirical justification for why MorphKV should outperforms existing KV compression methods.

Additionally, since the KV cache size is a hyperparameter, it is unclear how to determine its optimal value in practice while ensuring performance preservation.

**Essential References Not Discussed:**

I do not find any essential references that need discussion.

**Experimental Designs Or Analyses:**

Please check Methods And Evaluation Criteria above.

**Methods And Evaluation Criteria:**

The evaluation criteria overall make sense.

However, I found that the full attention model performs worse than KV cache pruning methods, including MorphKV, in LongWriter and LongGenBench. The authors do not provide sufficient explanation for these abnormal results. (In contrast, in LongBench, one of the most widely used benchmarks, the full attention model maintains top performance.)

The paper primarily presents overall benchmark performances, which does not help in gaining a fine-grained understanding of the method. For better clarity, the authors could include example-level or task-level (e.g., retrieval, reasoning) analyses to offer deeper insights into their approach.

**Other Comments Or Suggestions:**

In Figure 3, it appears the authors have **misunderstood the H2O baseline**. This method dynamically evicts tokens from the KV cache during generation, rather than retaining all early tokens. As presented, Figure 3 is misleading. (For reference, please see their paper: https://arxiv.org/pdf/2306.14048)

**Other Strengths And Weaknesses:**

Strengths
- The paper is generally easy to read.
- The paper is self-contained.

Weaknesses
- [**Important**] The paper does not include an analysis of inference time. A key issue is that the proposed method is not compatible with fused attention kernels, such as FlashAttention, since it requires attention scores at each decoding step, which are unavailable for fused kernels.
- The approach appears to be a combination of SnapKV (prefilling phase) and H2O (decoding phase). I find little novelty or new insights in the proposed method.

**Questions For Authors:**

- Could you provide statistical testing on LongWriter and LongGenBench? I am not convinced that the full attention model performs worse than KV cache pruning methods. I suspect there may be significant variance in performance.
- Is MorphKV compatible with FlashAttention?

**Relation To Broader Scientific Literature:**

This paper's evaluation is limited to NLP, but the inference algorithm could also be applied to Transformer models in other domains.

**Theoretical Claims:**

The paper does not contain theoretical claims.

---

> ### Author Rebuttal · Authors · 2025-04-01
>
> We thank the reviewer for their feedback.
>
> **Q-1) Comparison with Prior Works**
>
> R-1) MorphKV’s superior performance mainly stems from its more accurate token selection policy.
>
> StreamingLLM retains the KVs of the first few initial tokens called *attention sinks* and a sliding window of tokens, evicting all intermediate tokens between them. This creates a “context gap”, leading to partial capture of distant context as noted by prior works [1]. In contrast, MorphKV retains distant tokens (including important intermediate tokens) actively attended by recent tokens, enabling essential context capture throughout token generation.
>
> MorphKV eliminates the token selection bias dominant in H$_2$O and Keyformer. MorphKV exploits the insight that due to the auto-regressive nature of token generation, each token attends to all its past tokens and thus, is naturally aware of useful distant tokens. MorphKV retains only those distant tokens identified as relevant by preceding recent tokens, allowing the current output token to focus only on a curated set of useful distant tokens.
>
> SnapKV assumes that critical tokens from the input prompt remain critical throughout token generation. Based on this insight, it does not prune the KV cache during token generation. However, LLMs often suffer from *degeneration* while generating large responses [2] as each token attends to all past tokens, compounding noise from early decoding steps. As SnapKV retains KVs of all generated tokens, it is significantly more vulnerable to degeneration and performance degradation for long-response tasks. In contrast, MorphKV only retains *important* tokens in the KV cache enabling it to generate more coherent responses.
>
> Degeneration is quantified using the rate at which phrases are repeated in the LLM response. **Table-9** shows that SnapKV has 1.3$\times$ higher repetition rate than MorphKV on LongWriter.
>
> ### **Table 9**: Text Degeneration via N-Gram Repetition
> |Llama3.1-8B-Instruct|MorphKV|SnapKV|Full-Attention|
> |-|-|-|-|
> |Repetition Rate|68%|89%|89%|
>
> [1] Attention-Gate: https://arxiv.org/pdf/2410.12876
>
> [2] Text Degeneration: https://arxiv.org/abs/1904.09751
>
> **Q-2) Why MorphKV outperforms Full-Attention (FA)**
>
> R-2) Although we intuitively expect FA to perform superior to KV cache pruning techniques, for the same reasons as described above (degeneration), it may actually perform worse for certain tasks. Recent works, like SnapKV, and PyramidKV also make similar observations. SnapKV argues that FA often introduces noise to the attention mechanism by retaining all tokens, hindering its ability to attend to only the most relevant tokens more strongly. This trend is heavily task-dependent and KV cache compression does not always guarantee outperforming full-attention. For example, while MorphKV outperforms FA for few long-response tasks, FA outperforms MorphKV by 1.8% on LongGenBench for Qwen2.5-32B-Instruct.
>
> **Q-3) Statistical testing**
>
> R-3) Our results do not exhibit any statistical variance because MorphKV performs greedy decoding, and therefore generated responses remain same across runs.
>
> **Q-4) Optimal Value of KV cache size**
>
> R-4) We can obtain the optimal values through Grid-Search with validation on the end task. For our evaluations, we perform a coarse-grained search and determine that a KV cache size of 4K tokens is a practical starting point for most tasks.
>
> **Q-5) Generalizability of MorphKV**
>
> R-5) Please refer to **R-4** in response to **Reviewer vs1e**.
>
> **Q-6) Inference Time**
>
> R-6) Please refer to **R-5** in response to **Reviewer Jsuc**.
>
> **Q-7) Comparison of MorphKV against combination of SnapKV and H$_2$O**
>
> R-7) MorphKV differs substantially from a naive combination of SnapKV and H$_2$O. This primarily stems from the token selection policy used in MorphKV. Although both H$_2$O and MorphKV prune the KV cache dynamically during the decode phase, they use different policies to identify the *important* KVs they must retain. Consequently, even for the same memory budget, the set of KVs retained by the two approaches is significantly different. Please refer to R-1 for a detailed discussion regarding how MorphKV compares with SnapKV.
>
> **Q-8) H$_2$O Illustration**
>
> R-8) We use Figure 3 only as an illustrative example to highlight that H$_2$O’s token retention policy introduces a bias towards early tokens due to aggregated attention scores, similar to *Figure 2 (a)* in NACL [1]. We will update the figure to highlight the dynamic token-eviction in H$_2$O.
>
> [1] NACL: https://arxiv.org/pdf/2408.03675
>
> **Q-9) Compatibility with FlashAttention**
>
> R-9) Yes, MorphKV is already compatible with FlashAttention (please see **Section 4** of the paper). While fused kernels do not provide attention scores directly, It is still possible to tweak the FlashAttention kernel to return partial attention matrices alongside the final attention output, which can be consumed by MorphKV. An in-depth discussion of this is beyond the focus of this work.

---

> > ### Comment · Reviewer_FtVG · 2025-04-04
> >
> > I appreciate the detailed rebuttal! I understand your point regarding the H2O description. To avoid confusion and clearly convey the message, I hope the authors update the figure.
> >
> > Regarding Table 9, which is informative, more fine-grained analyses that provide insights into the working mechanisms of the proposed methods would be valuable for improving the paper.
> >
> > Regarding fused attention, you might use FlashAttention alongside but fusing your method inside the fused algorithm is not trivial, as the FlashAttention kernel does not calculate partial attention matrices. It only maintains blocks of **output** features, which are recurrently updated using the partial **unnormalized attention score** (no softmax over entire keys) and scaling factors. Storing the full attention score per query token in GPU SRAM is infeasible for long-context, so your approach might decrease FlashAttention performance even if integrated.  There are studies highlighting post-training compression methods do not achieve theoretical improvements in real-world applications due to a lack of hardware awareness [1].
> >
> > Despite these limitations, the authors have clarified my questions, so I have updated my score to weak reject.
> >
> > [1] Yuan et al., Native Sparse Attention: Hardware-Aligned and Natively Trainable Sparse Attention, 2025

---

> > > ### Author Response · Authors · 2025-04-07
> > >
> > > We sincerely thank the reviewer for their thoughtful follow-up and updated score. We will revise our manuscript to include Table 9. We are in the process of developing a modified FlashAttention kernel in Triton. MorphKV mitigates storage overhead by using attention profiles only from recent window tokens. However, as even these may exceed SRAM capacity, we offload partial unnormalized attention blocks to HBM, albeit at the cost of increased memory bandwidth.
> > >
> > > We thoroughly appreciate the comments related to fused kernels and will highlight the associated challenges and trade-offs in the final version of our paper.

---

### Official Review · Reviewer_Jsuc · 2025-03-24

**Overall Recommendation:** 3

**Summary:**

This paper introduces **MorphKV**, a method that dynamically selects caching tokens in pre-trained language models during inference. Unlike prior approaches such as **streamingLLM** and **SnapKV**, MorphKV employs two metrics—*sum fusion* and *max fusion*—to identify and retain tokens most closely attended to by recent tokens during both the **prefill** and **decoding** stages. By leveraging this dynamic caching strategy, MorphKV effectively supports generation tasks involving long contexts and extended responses.

**Claims And Evidence:**

The paper is well-written, and most of the claims are supported by evidence.
In lines 381-384, “this suggests that larger models can better leverage MorphKV…” seems to be not supported by existing experiments.

**Essential References Not Discussed:**

Not applicable

**Experimental Designs Or Analyses:**

In the paper, the author conducted experiments in both long response and long context scenarios.

**Methods And Evaluation Criteria:**

Standard benchmark like LongBench is utilized in the paper for evaluation. Also proposed a long-response scenario for evaluation.

**Other Comments Or Suggestions:**

N/A.

**Other Strengths And Weaknesses:**

### Strengths

1. MorphKV addresses an important challenge in LLM inference, particularly relevant to scenarios involving long-context processing and long-response generation.
2. The paper is well-organized, clearly written, and easy to follow.

### Weaknesses

1. The largest model evaluated in the experiments is Phi-4 (14B parameters). The authors' claim in lines 381–384 that "larger models can better leverage MorphKV" is speculative and not sufficiently supported by their current experimental results.
2. MorphKV appears to be a relatively minor modification to existing caching methods. The novelty could be better highlighted by clearly differentiating it from previous methods, such as streamingLLM and SnapKV.
3. Long-context and long-response generation present distinct caching challenges. The paper lacks detailed analysis or justification for why these particular tasks are the most appropriate to demonstrate the efficacy of the proposed strategy.

**Questions For Authors:**

1. In the Appendix, only an ablation regarding the number of recent tokens and fusion strategy is presented. Do you expect performance differences when varying the total number of cached tokens beyond just recent tokens?
2. Dynamically selecting tokens for caching appears computationally expensive. Could you clarify whether this is indeed the case? If so, could you provide runtime comparisons between MorphKV, SnapKV, and H2O?
3. Previous methods typically retain older tokens and their all associated information (i.e., full column). In MorphKV, are initial tokens progressively evicted as generation length increases? If tokens are evicted, could this negatively impact performance in tasks that heavily rely on information retrieval?

**Relation To Broader Scientific Literature:**

In the paper, there are two major concepts that are prefilled token selection and decode token selection. For prefil,l it is closely related to SnapKV, while MorphKV utilizes slightly different metrics. For decode head, it is more closely related to H2O.

**Theoretical Claims:**

Not applicable.

---

> ### Author Rebuttal · Authors · 2025-04-01
>
> We thank the reviewer for their feedback. We have addressed the points raised in the review below:
>
> **Q-1) Analysis on Larger Models**
>
> R1) MorphKV remains effective for larger models, as demonstrated by our recent evaluations with larger models. We request the reviewer consult **Table-1** and **Table-2** in the response to **Reviewer fibk**.
>
> **Q-2) Comparison with Prior Works**
>
> R2) For a detailed comparison with prior works, including StreamingLLM and SnapKV, please refer to **R-1** in response to **Reviewer-FtVG**.
>
> **Q-3) Choice of Benchmarks**
>
> R-3) Both long-context and long-response generation tasks cater to widely prevalent applications of LLMS and are key areas of focus [1-3]. Long-context generation not only represents common use cases of LLMs like text summarization and passage retrieval but can also be used for innovative training techniques such as multi-shot learning, where the model can learn from hundreds of training examples provided directly in the prompt.
>
> In contrast, long-response generation is crucial for tasks such as story generation, paragraph completion, comprehensive question-answering, content creation, etc. While prior works, such as SnapKV, are optimized explicitly for long-context tasks, they are inefficient for long-response tasks. MorphKV addresses this critical bottleneck.
>
> [1] Google: https://cloud.google.com/transform/the-prompt-what-are-long-context-windows-and-why-do-they-matter
>
> [2] IBM: https://research.ibm.com/blog/larger-context-window
>
> [3] Amazon: https://aws.amazon.com/blogs/security/context-window-overflow-breaking-the-barrier/
>
>
> **Q-4) Effect of KV cache budget on MorphKV**
>
> R-4) Thanks for the detailed feedback. We have conducted additional studies using LongGenBench tasks for the Mistral-24B-Instruct [1] model to evaluate the impact of varying cache capacities on performance. We observe that the performance of both MorphKV and H$_2$O increases with the number of cached tokens. At the same KV cache capacity, MorphKV remains more effective than H$_2$O. We will revise the Appendix to include these results.
>
> ### **Table 7**: Performance with increasing KV cache capacity on LongGenBench
> |KV cache compression method (number of cached tokens) |Completion Rate|Average Accuracy|
> |-|-|-|
> |H$_2$O (1000)  | 56.46% | 35.0%|
> |H$_2$O (2000)  | 61.35% | 52.0%|
> |H$_2$O (4000)  | 61.35% | 58.4%|
> |MorphKV (1000) | 61.64% | 40.0%|
> |MorphKV (2000) | 61.64% | 54.0%|
> |MorphKV (4000) | 61.64% | 58.4%|
>
> [1] Mistral-24B-Instruct: https://mistral.ai/news/mistral-small-3
>
> **Q-5) Is dynamic token selection for caching computationally expensive? If so, how does the runtime of MorphKV compare to SnapKV and H$_2$O?**
>
> R-5) KV cache compression methods must navigate complex trade-offs encompassing accuracy, inference time, throughput, and memory footprint. Optimizing for a single metric alone is insufficient for practical adoption.
>
> MorphKV prunes the cache at every token processing step, which increases the computational overhead. Our implementation employs several optimizations to reduce these overheads, such as CPU offloading and prefetching with dedicated CUDA streams that minimize the time overhead associated with loading attention weights.
>
> Compared to a similar dynamic token eviction policy, such as H$_2$O, MorphKV achieves **8%** faster inference time (while remaining more accurate). MorphKV’s runtime is higher than SnapKV’s, which is expected because SnapKV is tailored for long-context tasks, not long-response ones.
>
> However, the reduced memory footprint of the KV caches enables MorphKV to deliver a much higher throughput (up to 4.68$\times$) due to a larger batch size, effectively compensating for the degradation observed in the runtime of each request. In conclusion, MorphKV outperforms existing KV cache compression techniques by achieving the best overall balance: it reduces KV cache memory usage by up to 85%, increases throughput by 4.68$\times$, and achieves an inference time improvement of 8% over state-of-the-art methods (such as H$_2$O) – all while preserving or improving accuracy. **Table 8** summarizes these results.
>
> ### **Table 8**: Comparison of key metrics across KV cache compression methods for Mistral-7B on LongBench
> |Task|SnapKV|H$_2$O|MorphKV|
> |-|-|-|-|
> | Runtime | 1$\times$ |  1.62$\times$ | 1.50$\times$ |
> | Throughput | 1$\times$ | 2.4$\times$ | 4.68$\times$ |
> | Accuracy | 1$\times$ |  0.94$\times$ | 1.01$\times$ |
> | Memory | 1$\times$ | 0.26$\times$ | 0.14$\times$ |
>
> **Q-6) MorphKV on Retrieval Tasks**
>
> R-6) MorphKV dynamically evicts tokens based on attention weight scores of the recent window tokens. Hence, it is not biased towards retention or eviction of initial tokens during generation. We evaluate MorphKV on the LongBench suite, which includes many retrieval tasks (single-doc QA, multi-doc QA, Passage Retrieval etc.), and MorphKV shows robust performance across all these tasks.

---

> > ### Comment · Reviewer_Jsuc · 2025-04-03
> >
> > I would like to thank the authors for their detailed rebuttal! Their supplementary results have further demonstrated the effectiveness of the algorithm! I will stand with my current score.

---

> > > ### Author Response · Authors · 2025-04-04
> > >
> > > We sincerely appreciate the reviewer’s time and positive feedback. Thank you for acknowledging our supplementary results and for your support!

---

### Official Review · Reviewer_vs1e · 2025-03-25

**Overall Recommendation:** 3

**Summary:**

The paper introduces MorphKV, a novel method for efficiently managing key-value (KV) caches in Large Language Models (LLMs) while maintaining memory efficiency and model accuracy. The method overcomes the problem of growing memory requirements for KV caches during inference by employing a dynamic, correlation-aware token selection process. MorphKV reduces memory usage by up to 88.1%, providing higher accuracy and scalability compared to current state-of-the-art methods such as SnapKV and H2O.

**Claims And Evidence:**

MorphKV claims to offer substantial improvements in both memory efficiency and model performance. Through experimental evaluations on various LLMs (e.g., Llama, Mistral, Qwen), the authors show that MorphKV achieves 86.5% memory savings with a 13.5% increase in accuracy over previous methods. These results are demonstrated across multiple benchmarks, including long-response generation and long-context understanding tasks.

**Essential References Not Discussed:**

While the paper cites numerous relevant works, it could have explored more recent advancements in KV cache optimization and memory-efficient model architectures. For example, methods that combine KV cache compression with layer-specific optimizations might provide additional insights for further improvements. (i.e. PyramidKV, Ada-KV, HeadKV)

**Experimental Designs Or Analyses:**

The experimental design includes comparative studies using different state-of-the-art methods for KV cache compression. The results indicate that MorphKV performs favorably in both memory efficiency and task accuracy. The experiments are robust, spanning a variety of LLMs and task categories, and also consider different fusion strategies (sum and max fusion) for selecting KV tokens.

**Methods And Evaluation Criteria:**

The authors evaluate MorphKV's performance on tasks such as long-response generation (LongWriter), long-context understanding (LongBench), and structured long-response tasks (LongGenBench). These tasks are compared to SnapKV and H2O, using key metrics like model accuracy, KV cache sizes, and task completion rates. The method's memory efficiency is also tested under varying response lengths, with MorphKV demonstrating stability even as outputs grow larger.

**Other Comments Or Suggestions:**

Further work could explore the integration of MorphKV with other model architectures, particularly those that focus on multi-modal data. The efficiency of MorphKV could also be tested on more specialized benchmarks, such as those involving medical or legal text generation, to assess its generalizability across domains. Additionally, future versions could investigate the computational overhead introduced by the dynamic KV cache selection process.

**Other Strengths And Weaknesses:**

Strengths: The paper proposes a practical, real-time solution for LLM inference tasks that need memory efficiency without sacrificing performance. MorphKV's ability to scale with response length is a notable advantage for tasks involving extended outputs, which are increasingly common in applications like content generation and interactive assistants.

Weaknesses: The paper could have benefited from more detailed discussions on the potential trade-offs between the computational complexity of the token selection process and its memory efficiency. The impact of MorphKV on inference speed is not discussed in detail, which could be a critical factor in real-time applications.

**Questions For Authors:**

No

**Relation To Broader Scientific Literature:**

The work builds on existing research in KV cache compression for LLMs, extending previous approaches like SnapKV, H2O, and Keyformer. By dynamically selecting relevant tokens, MorphKV advances these methods, offering better scalability and efficiency. The authors also relate their findings to memory management strategies in other machine learning models and systems.

**Theoretical Claims:**

The paper introduces the theory that the selective retention of KV tokens based on correlation with recent tokens helps to preserve long-range dependencies while minimizing memory usage. This claim is supported by the experimental results, where MorphKV's selective KV cache management outperforms prior methods that either discard or retain uncorrelated tokens.

---

> ### Author Rebuttal · Authors · 2025-04-01
>
> We thank the reviewer for their appreciation of our work. We have addressed the point raised in the review below:
>
> **Q-1) Paper Could Explore PyramidKV, Ada-KV, HeadKV Advancements**
>
> R-1) Thank you for the excellent suggestion. MorphKV is orthogonal to methods like PyramidKV, which optimizes the KV cache across layers, and Ada-KV and Head-KV, which optimize the KV cache across attention heads. Thus, integrating MorphKV with these methods can improve its efficacy even further. In fact, we have already conducted some preliminary studies wherein we simulate a layer-wise optimization by naively deactivating MorphKV for the first three layers based on prior studies that describe the importance of information in the early layers [1,2,3]. Our studies show that this design is even more effective (with roughly 10% additional memory requirements) than the default implementation of MorphKV, which prunes the KV cache uniformly across all model layers. The results are summarized in **Table 5** below.
>
> Integrating MorphKV with methods like PyramidKV, Ada-KV, and Head-KV as a generalized solution requires a deeper understanding of how the methods complement each other, careful tuning of the design parameters, and development of a robust policy that dynamically combines them for optimal performance. Such an extensive study is beyond the scope of our current paper, and we reserve it for future work.
>
> ### **Table 5**: MorphKV with basic layer-wise optimization for Mistral-7B on LongGenBench
> |Task|Completion Rate|Average Accuracy|
> |-|-|-|
> |MorphKV (default) | 70.51 | 44.2%|
> |MorphKV (layer-wise optimization) | 71.96 (2.05% better) | 46.1% (4.29% better)|
>
> MorphKV configuration has a capacity of 2000 tokens
>
> [1] PyramidKV: https://arxiv.org/pdf/2406.02069
>
> [2] SqueezeAttention: https://openreview.net/forum?id=9HK2rHNAhd
>
> [3] Layer-condensed KV cache: https://aclanthology.org/2024.acl-long.602.pdf
>
> **Q-2) Trade-Off between Computational Complexity and Memory Efficiency**
>
> R-2) By default, MorphKV prunes the KV cache after each token is processed to attain a constant-sized cache, which incurs additional computational overheads. The overheads can indeed be lowered by adopting a *lazy* token-selection policy, at the cost of slight performance degradation. The computational savings scale in the number of pruning steps skipped by the lazy token-selection policy. For example, if MorphKV prunes the cache after every 10 tokens processed, the computational overheads reduce to 0.1$\times$ of that required in the default MorphKV.
>
> To study this further, we implement a variant of MorphKV, which allows the KV cache to exceed the pre-allocated capacity before pruning it back. **Table 6** compares the performance of the default and lazy variant of MorphKV. We will include these results in the revised manuscript.
>
> ### **Table 6**: MorphKV with lazy token-selection for Mistral-7B on LongGenBench
> |Task|Completion Rate|Average Accuracy|Runtime|
> |-|-|-|-|
> |MorphKV (default) | 72.10 | 47.0%|1$\times$|
> |MorphKV (with lazy token-selection) | 71.86 (0.34% worse) |46.1% (1.9% worse)|0.82$\times$|
>
> MorphKV configuration has a capacity of 4000 tokens
>
> **Q-3) Inference Time**
>
> R-3) We request the reviewer to consult R-5 in response to **Reviewer Jsuc**.
>
> **Q-4) Generalizability of MorphKV**
>
> R-4) We already evaluate MorphKV using the LongBench [1] suite that encompasses tasks such as question answering [2,3] based on academic, legal, government, literature and financial reports, reasoning [4-6] from articles and reports from wikipedia, and summarization [7] on documents from academia and product development (please see **Section-5.3**). MorphKV delivers robust performance across these tasks, showing its generalizability across various domains
>
> [1] LongBench: https://arxiv.org/pdf/2308.14508
>
> [2] NarrativeQA: https://ar5iv.labs.arxiv.org/html/1712.07040
>
> [3] Qasper: https://paperswithcode.com/dataset/qasper
>
> [4] HotpotQA: https://paperswithcode.com/dataset/hotpotqa
>
> [5] 2wikiMQA: https://aclanthology.org/2020.coling-main.580/
>
> [6] PassageCount: https://arxiv.org/pdf/2308.14508
>
> [7] QMSum, VCsum: https://arxiv.org/pdf/2308.14508

---

### Official Review · Reviewer_bnHa · 2025-03-25

**Overall Recommendation:** 4

**Summary:**

This work designed and developed an efficient KV cache management technique to keep constant KV size while achieving higher accuracy  for long context and long response tasks. The author has compared with relevant works, e.g., SnapKV, H2O and full attention, etc, using different model and different benchmarks comprehensively. The evaluation shows that MorphKV achieved start of art performance on KV size controlling and benchmark task accuracy.

**Claims And Evidence:**

Yes. The claims and evidence are clear and well backed by numbers and detailed step by step illustration.

**Essential References Not Discussed:**

Yes.

**Experimental Designs Or Analyses:**

Yes. The experimental design and analysis are done on a H200 with NVLink.

**Methods And Evaluation Criteria:**

Yes. The method makes sense to the problem, i.e., keeping relevant context in the older tokens is helpful for the performance.
The evaluation process is comprehensive, i.e., different models and benchmarks are evaluated to compare MorphKV with existing works.

**Other Comments Or Suggestions:**

NA

**Other Strengths And Weaknesses:**

NA

**Questions For Authors:**

How easy to use in real world system.

**Relation To Broader Scientific Literature:**

Yes. The key contribution of this paper is it identified a better attention mechanism related to the coherence and semantic meaning of all the tokens, i.e., history token and recent token. Such that, more research can be inspired from this work on how to retain older tokens for better performance and optimized KV cache management.

**Theoretical Claims:**

Yes, the theory and math are correct. The key part is to calculate the attention weights, which makes sense and seems correct.

---

> ### Author Rebuttal · Authors · 2025-04-01
>
> We thank the reviewer for their feedback. We have addressed the points raised in the review below:
>
> **Q-1) Effect of Window Size on MorphKV**
>
> R-1) The window size indeed affects the performance of MorphKV. In practice, **all** KV cache pruning methods rely on hyperparameters: examples include total cache capacity in H$_2$O and Keyformer, window sizes in StreamingLLM and SnapKV, and the number of layers for KV cache projection in SwiftKV. Careful tuning of these parameters is essential for practical adoption and achieving substantial memory savings [1,2].
>
> Similarly, MorphKV also benefits from tuning its window size parameter. Our experiments show that a window size of 32 consistently offers high performance across all LongBench suite tasks, including diverse scenarios such as code generation and literature review. Although increasing window sizes can slightly improve performance for tasks requiring information recall over longer spans, it typically yields diminishing returns, as described in **Table 3** and **Table 4**. Hence, the window size can be coarsely approximated based on the nature of the task in terms of the information span to achieve good performance.
>
>
>
> ### **Table 3**: Mistral-24B-Instruct performance on LongGenBench
> | MorphKV Configuration | Completion Rate | Average Accuracy |
> |:--:|:--:|:--:|
> | window size: 32 | 61.55%  | 54.3% |
> | window size: 200 | 61.64%  | 58.4% |
>
> ### **Table 4**: Qwen2.5-32B-Instruct performance on LongGenBench
> | MorphKV Configuration | Completion Rate | Average Accuracy |
> |:--:|:--:|:--:|
> | window size: 32 | 71.68%  | 51% |
> | window size: 200 | 71.68%  | 53% |
>
> NOTE: each scenario above has a total KV cache capacity of 4000 tokens
>
> [1] Accelerating Enterprise LLM Workloads: https://www.snowflake.com/en/engineering-blog/swiftkv-llm-compute-reduction/
>
> [2] StreamingLLM Integration into TensorRT-LLM: https://github.com/NVIDIA/TensorRT-LLM/tree/main/examples/llama#run-llama-with-streamingllm
>
> **Q-2) Ease of Usage in Real-World Systems**
>
> R-2) Currently, MorphKV is implemented in the widely used transformers library from HuggingFace (see **setup in Section-4**). MorphKV is integrated inside both standard Attention and FlashAttention modules and uses cache offloading to accommodate demanding memory-intensive use cases. Consequently, MorphKV is compatible with a vast spectrum of models hosted on HuggingFace. Our approach is consistent with prior works and facilitates seamless adoption across LLMs for real-world scenarios.

---

### Official Review · Reviewer_fibk · 2025-03-26

**Overall Recommendation:** 4

**Summary:**

This paper presents MorphKV to reduce the KV cache in long LLM context. Its dynamic KV selection algorithm improves the accuracy by 18.6 and 13.6 compared to previous SnapKV and H2o, while reducing KV by 88.1 and 51.6.

## update after rebuttal
During the rebuttal, the authors add experiments on larger (24B, 32B) to demonstrate the effectiveness of the method. Though I did not further increase my score (my initial score is 4), I am very supportive of the proposed methods and paper.

**Claims And Evidence:**

The paper claims to achieve lower KV cache usage and higher accuracy, which is well supported by Table 1 and Figure 6.

**Essential References Not Discussed:**

The paper discusses the majority of related works.

**Experimental Designs Or Analyses:**

The experimental designs make sense. It evaluates against the two baselines and the above mentioned LLMs and benchmarks. It shows the method reduces the KV cache size with improved accuracy.

**Methods And Evaluation Criteria:**

Yes, it runs on llama3-8b, Mistral-7b, qwen-7b, phi 14b which are leading LLMs. It evaluates on LongWriter, LongBench; it compares against SnapKV and H2o, which all make sense.

**Other Comments Or Suggestions:**

I am wondering whether the author can include more analysis on larger models. The current models are 7-14B, where the efficiency issue in larger models is more severe. However, the reviewer votes for accept due to the clear effectiveness of the method.

**Other Strengths And Weaknesses:**

The paper is well written, and the results are significant.

**Questions For Authors:**

Please see the above comments.

**Relation To Broader Scientific Literature:**

Related work in compressing KV are either not scalable or has limited accuracy (as discussed in the background section). This paper introduces a scalable method with high accuracy after KV compression.

**Theoretical Claims:**

There are no theoretical claims.

---

> ### Author Rebuttal · Authors · 2025-04-01
>
> We appreciate the reviewer's feedback on both the structure of our paper and the results. We have addressed the points raised in the review below:
>
> **Q-1) Analysis on Larger Models**
>
> R-1) MorphKV remains effective even for larger models. We evaluate MorphKV using long-response tasks from the LongGenBench suite on an Nvidia H100 GPU equipped with 80 GB of HBM-2e memory on two models (Mistral-24B-Instruct [1] and Qwen2.5-32B-Instruct [2]). Even for a single task, SnapKV consumes significant KV cache sizes: 6.16 GB (50.26 GB including weights) and 9.7 GB (71.14 GB including weights), respectively, nearly exhausting the GPU's available HBM capacity. In contrast, MorphKV requires only 0.80 GB and 1.06 GB, respectively. The tables below summarize these results. The evaluated model sizes are consistent with prior works, like SnapKV (7B), H$_2$O (30B), and Keyformer (7B).
> ### **Table-1**: Mistral-24B-Instruct on LongGenBench
> | KV Cache Compression Method | Completion Rate | Average Accuracy |  KV cache size  |
> |:--:|:--:|:--:|:--:|
> | SnapKV  |  61.40% | 57.8% | 7$\times$ |
> | H$_2$O  |  61.35% | 58.4% | 3.6$\times$ |
> | **MorphKV** | **61.64%** | **58.4%** | **1$\times$** |
>
> ### **Table-2**: Qwen2.5-32B-Instruct on LongGenBench
> | KV Cache Compression Method | Completion Rate | Average Accuracy |  KV cache size  |
> |:--:|:--:|:--:|:--:|
> | SnapKV |  71.59% | **54%** | 9.1$\times$ |
> | H$_2$O |  71.39% | 53% | 4.6$\times$ |
> | **MorphKV** | **71.68%**  | 53% | **1$\times$** |
>
> [1] Mistral-24B-Instruct: https://mistral.ai/news/mistral-small-3
>
> [2] Qwen2.5-32B-Instruct: https://qwenlm.github.io/blog/qwen2.5/

---

> > ### Comment · Reviewer_fibk · 2025-04-02
> >
> > Thank you for getting back! I maintain my score and support this paper.

---

> > > ### Author Response · Authors · 2025-04-04
> > >
> > > Thank you so much for supporting our paper! We sincerely appreciate your time and thoughtful feedback in reviewing it.

---

### Decision · Program_Chairs · 2025-05-01

**Decision:**

Accept (poster)

**Comment:**

**Summary:**
This paper presents MorphKV, a dynamic KV cache management method that reduces memory usage during inference in large language models by selectively retaining high-utility tokens based on attention patterns. The method demonstrates strong results across multiple benchmarks and models, achieving notable memory savings while maintaining or improving accuracy, though questions remain regarding novelty, inference efficiency, and compatibility with fused attention implementations.

**Pros:**
1. Proposes a practical and effective KV cache compression technique that improves upon prior methods (e.g., SnapKV, H2O) in both memory efficiency and task accuracy.
2. Empirically validated on multiple models (up to 14B) and long-context/long-response benchmarks, with ablation studies included.
3. Well-written, well-organized, and shows potential for scaling to more complex inference scenarios.

**Cons:**
Some reviewers questioned the novelty of the approach, raised concerns about its compatibility with fused attention and computational overhead, and noted limited evaluation on larger models and diverse long-context tasks.